# Insulin-producing β-cells regenerate ectopically from a mesodermal origin under the perturbation of hemato-endothelial specification

Ka-Cheuk Liu[1], Alethia Villasenor[2], Maria Bertuzzi[3], Nicole Schmitner[1], Niki Radros[4], Linn Rautio[1], Kenny Mattonet[2], Ryota L Matsuoka[2,5], Sven Reischauer[2,6], Didier YR Stainier[2], Olov Andersson[1]*

[1]Department of Cell and Molecular Biology, Karolinska Institutet, Stockholm, Sweden; [2]Department of Developmental Genetics, Max Planck Institute for Heart and Lung Research, Bad Nauheim, Germany; [3]Department of Neuroscience, Karolinska Institutet, Stockholm, Sweden; [4]Dermatology and Venereology Division, Department of Medicine (Solna), Karolinska Institutet, Stockholm, Sweden; [5]Department of Cardiovascular and Metabolic Sciences, Lerner Research Institute, Cleveland Clinic, Cleveland, United States; [6]Cardio-Pulmonary Institute, Frankfurt, Germany; Medical Clinic I, (Cardiology/Angiology) and Campus Kerckhoff, Justus-Liebig-University Giessen, Giessen, Germany

*For correspondence:
Olov.Andersson@ki.se

Competing interest: See
page 19

Reviewing editor: Elke Ober,
University of Copenhagen,
Denmark

**Abstract** To investigate the role of the vasculature in pancreatic β-cell regeneration, we crossed a zebrafish β-cell ablation model into the avascular *npas4l* mutant (i.e. *cloche*). Surprisingly, β-cell regeneration increased markedly in *npas4l* mutants owing to the ectopic differentiation of β-cells in the mesenchyme, a phenotype not previously reported in any models. The ectopic β-cells expressed endocrine markers of pancreatic β-cells, and also responded to glucose with increased calcium influx. Through lineage tracing, we determined that the vast majority of these ectopic β-cells has a mesodermal origin. Notably, ectopic β-cells were found in *npas4l* mutants as well as following knockdown of the endothelial/myeloid determinant Etsrp. Together, these data indicate that under the perturbation of endothelial/myeloid specification, mesodermal cells possess a remarkable plasticity enabling them to form β-cells, which are normally endodermal in origin. Understanding the restriction of this differentiation plasticity will help exploit an alternative source for β-cell regeneration.

## Introduction

The concept of embryonic development and cell fate determination was illustrated by the famous Waddington landscape model decades ago (*Waddington, 1957*). Waddington's model not only shows the importance of spatiotemporal precision in cell differentiation but also metaphorises cell fate determination as a sequential and irreversible event. In this hierarchical model, the endoderm follows the lineage paths downwards and progressively differentiates into multiple endodermal cell types, including pancreatic β-cells. Likewise, the mesoderm stays in the mesodermal lineage paths and differentiates into endothelial cells and other mesodermal cell types. However, in recent decades, multiple studies have suggested that committed cells are capable of differentiating across the germ layer boundaries by converting embryonic and/or adult mesodermal fibroblasts into ectodermal neuronal cells (*Vierbuchen et al., 2010*), multipotent neural stem cells (*Ring et al., 2012*),

endodermal hepatocyte-like cells (*Huang et al., 2011*; *Sekiya and Suzuki, 2011*), or pancreatic β-like cells (*Zhu et al., 2016*) in vitro. These studies highlight the feasibility of converting mesodermal cells into ectodermal or endodermal cells in vitro.

Despite the extensive studies on cell fate conversion across germ layers in vitro, the number of in vivo studies is limited. Ectopic expression of *Xsox17β* in *Xenopus* embryos relocated cells normally fated to become ectoderm to appear in the gut epithelium, suggesting a possible change in cell fate in vivo (*Clements and Woodland, 2000*). Similarly, ectopic expression of *sox32/casanova* in presumptive endothelial cells switched their cell fate to endoderm in zebrafish embryos (*Kikuchi et al., 2001*). Furthermore, aggregated morulae and chimeric mouse embryos of β-catenin mutants provided evidence of precardiac mesoderm formation in endodermal tissues in vivo (*Lickert et al., 2002*). These studies suggest that the classical in vivo germ layer boundaries may not be as clear-cut as previously thought.

In this study, we aimed to elucidate the importance of the vasculature in pancreatic β-cell regeneration, which plays a crucial role in potential therapeutic strategies against diabetes. We employed *cloche* zebrafish mutants as an avascular model. The mutation of *npas4l*, a master regulator of endothelial and hematopoietic cell fates, is responsible for the severe loss of most endothelial and blood cells in *cloche* mutants (*Parker and Stainier, 1999*; *Reischauer et al., 2016*; *Stainier et al., 1995*). Unexpectedly, the *npas4l* mutation induced ectopic β-cell formation in the mesenchymal region outside of the pancreas after β-cell ablation. Lineage-tracing mesodermal cells expressing *draculin* (*drl*), *npas4l*, and *etsrp* (previously named *etv2*) validated the mesodermal lineage of the ectopic β-cells, which are normally endodermal in origin. These findings offer novel insights into cell fate determination and an alternative source of β-cells.

## Results

### Ectopic β-cell formation in *npas4l* mutants

To determine the importance of vasculogenesis and vascularisation for β-cell regeneration, we examined β-cell formation in zebrafish carrying the *cloche* mutation (*npas4l*$^{s5}$ mutation) after β-cell ablation, i.e., in the *Tg(ins:Flag-NTR);Tg(ins:H2BGFP;ins:DsRed)* model. Nitroreductase (NTR), expressed under the *insulin* promoter, converts the prodrug metronidazole (MTZ) to a cytotoxin to specifically ablate insulin-producing β-cells (*Curado et al., 2007*). The homozygous mutation of *npas4l* significantly increased the number of *ins*:H2BGFP-positive cells during the β-cell regeneration period (*Figure 1A–C*). In addition, we observed a distinctive ectopic β-cell population in the mesenchymal region outside of the pancreas in the *npas4l*$^{−/−}$ group, an ectopic location that was very rarely observed in the sibling controls (including both wild-type and heterozygous siblings). This ectopic population of β-cells contributed to the major increase in the number of *ins*:H2BGFP-positive cells during β-cell regeneration (*Figure 1C*). Moreover, the comparable and sparse numbers of *ins*:DsRed-positive cells in the controls and mutants indicate that the *npas4l* mutation did not enhance the survival of β-cells during the ablation (*Figure 1A,B*) because the extended maturation time of DsRed (*Baird et al., 2000*) restricted the detection of DsRed to any surviving β-cells.

To visualise the location of the ectopic β-cells better, we labelled the pancreas with *ptf1a*:EGFP expression and observed not only a drastic reduction in the pancreas size (*Figure 1D,E*, *Figure 1—figure supplement 1*) but also the regeneration of β-cells clearly outside of the *ptf1a*-expressing exocrine pancreas in *npas4l* mutants (*Figure 1E*). By labelling the mesenchyme with *hand2*:EGFP expression (*Figure 1F–K*), we further revealed that the majority of the ectopic β-cells formed in *npas4l* mutants intermingled with *hand2*:EGFP-positive mesenchymal cells between the pronephros and the pancreas (*Figure 1J,K*). In addition, we occasionally observed ectopic β-cells intermingled with *hand2*:EGFP-positive mesenchymal cells ventral to the pancreas (*Figure 1I,K*). Although the ectopic β-cells were located among the mesenchymal cells, they did not express *hand2*:EGFP.

### The ectopic β-cells co-express insulin and endocrine markers in *npas4l* mutants

Next, we examined multiple pancreatic endocrine and β-cell markers, including Isl1, *neurod1*, *pdx1*, *mnx1*, *pcsk1*, and *ascl1b* (the functional homologue to *Neurog3* in mammals), to validate the β-cell identity of the ectopic insulin-producing cells. The majority of the ectopic β-cells co-expressed insulin

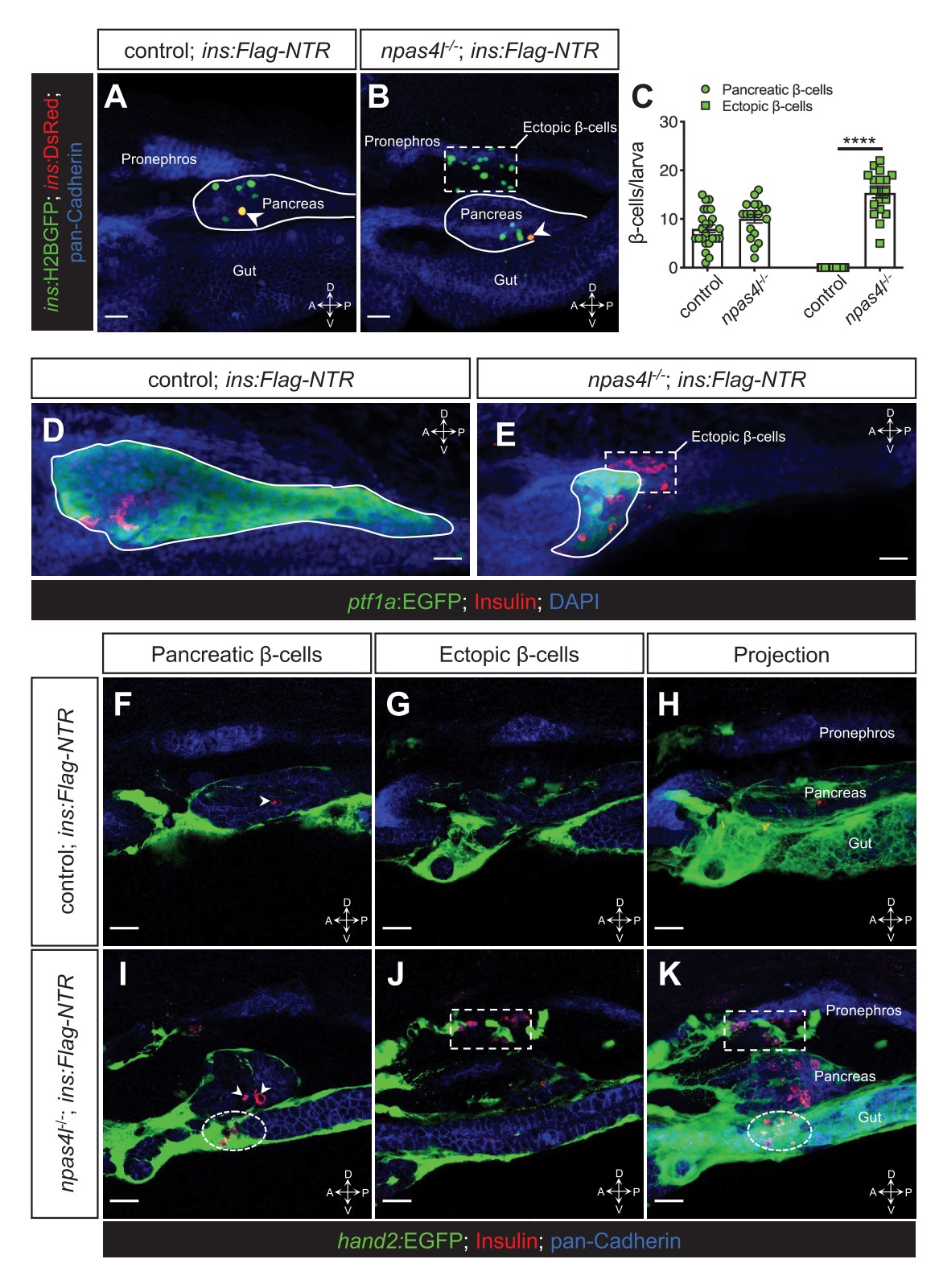

**Figure 1.** Ectopic β-cell formation in *npas4l* mutants. (**A, B**) Representative confocal projections of the pancreas and neighbouring tissues of control siblings and *npas4l*−/− Tg(ins:Flag-NTR);Tg(ins:H2BGFP;ins:DsRed) zebrafish larvae at 3 dpf after β-cell ablation by MTZ at 1–2 dpf, displaying regenerated β-cells in green and older β-cells that likely survived the ablation in yellow overlap as the DsRed fluorescence driven by the insulin promoter remained in these cells, at the same time as DsRed had not had enough time to mature in the regenerated β-cells (arrowheads). The ectopic

*Figure 1 continued on next page*

Figure 1 continued

β-cells are indicated by white dashed rectangle. Pancreata are outlined by solid white lines. (C) Quantification of the pancreatic or ectopic β-cells per larva at 3 dpf. ****p<0.0001 (Šidák's multiple comparisons test); n = 24 (control) and 19 (*npas4l*$^{-/-}$). Quantification data are represented as the mean ± SEM. (D, E) Representative image projections of the pancreas and neighbouring tissues in control siblings and *npas4l*$^{-/-}$ *Tg(ins:Flag-NTR);Tg(ptf1a: EGFP)* larvae at 3 dpf after β-cell ablation by MTZ at 1–2 dpf, displaying β-cells in red with immunostaining for insulin and *ptf1a*:EGFP$^+$ exocrine pancreas in green. Pancreata are outlined by solid white lines. Dashed line outlines ectopic β-cells in the mesenchyme (E). (F–K) Representative images and projections of the pancreas and neighbouring mesenchyme of control siblings and *npas4l*$^{-/-}$ *Tg(ins:Flag-NTR);Tg(hand2:EGFP)* zebrafish larvae at 3 dpf after β-cell ablation by MTZ at 1–2 dpf, displaying β-cells in red with immunostaining for insulin and *hand2*:EGFP$^+$ mesenchyme in green. White arrowheads point to β-cells in the pancreas (F, I). Dashed rectangles indicate the ectopic β-cells intermingling with the mesenchyme between the pronephros and the pancreas, without co-expressing insulin and *hand2*:EGFP (J, K). Selected area in dashed ovals shows other ectopic β-cells intermingling with the mesenchyme ventral to the pancreas (I, K). Scale bars = 20 µm. Anatomical axes: D (dorsal), V (ventral), A (anterior), and P (posterior).

The online version of this article includes the following figure supplement(s) for figure 1:

**Figure supplement 1.** Mutation of *npas4l* suppresses the development of exocrine pancreas.

as well as these markers during β-cell regeneration (*Figure 2*). The high co-expression of *pcsk1* (*Figure 2R,S*, *Table 1*), which encodes an enzyme necessary for insulin biosynthesis, indicates that most of the β-cells in the ectopic population are likely functional. Consistent with previous findings in pancreatic β-cells, not all ectopic β-cells expressed *ascl1b*:EGFP (*Figure 2V,W*, *Table 1*), which suggests that *ascl1b* works as a transient endocrine cell fate regulator (*Flasse et al., 2013*). In contrast with Isl1, *mnx1*, *pcsk1*, and *ascl1b*, we observed lower co-expression levels of *neurod1* and *pdx1* in ectopic β-cells compared with the pancreatic population in *npas4l* mutants (*Table 1*). In addition to the reduction in pancreas size (*Figure 1—figure supplement 1*), the *pdx1*-expressing pancreatic duct was also reduced in the *npas4l* mutant (*Figure 2—figure supplement 1*), indicating that the pancreas and its duct did not expand to form the ectopic β-cells. These observations together suggest that the pancreatic and ectopic β-cells are similar, yet they are two distinct β-cell populations.

## Ectopic β-cells can respond to glucose by displaying calcium oscillations

We further assessed the functionality and maturity of the ectopic β-cells by examining the calcium ion oscillation in vivo to determine their ability in responding to glucose to secrete insulin in the *Tg (ins:GCaMP6s);Tg(ins:mCherry);Tg(ins:Flag-NTR)* *npas4l* zebrafish mutants. After β-cell ablation by MTZ and one day of β-cell regeneration, we found comparable proportions of pancreatic (66%) and ectopic (67%) β-cells displaying mild and basal levels of calcium activity at the baseline (*Figure 3* and *Video 1*). Upon glucose treatment, 50% of pancreatic β-cells and 33% of ectopic β-cells elicited calcium oscillations, suggesting that a portion of the pancreatic and ectopic β-cells have the ability to respond to glucose to secrete insulin in the *npas4l* mutants during β-cell regeneration.

## The ectopic β-cells are of mesodermal origin in *npas4l* mutants and *etsrp* morphants

We have previously shown that *npas4l* expression is first initiated in the lateral plate mesoderm at the tailbud stage by in situ hybridisation (*Reischauer et al., 2016*). In this study, we examined *npas4l* expression at 20 hr postfertilisation (hpf) and found that *npas4l* was severely reduced in the lateral plate mesoderm in the *npas4l* mutants (*Figure 4—figure supplement 1A,B*), whereas normal expression levels were observed in the tailbud and brain. The cells with reduced *npas4l* expression were still present in the lateral plate mesoderm as demonstrated by the embryos incubated overnight to further develop the *npas4l* expression signal (*Figure 4—figure supplement 1B'*). On the contrary, there were no significant differences in the expression of the examined early mesodermal or endodermal transcription factors *foxa2*, *gsc*, and *mixl1*, or the pancreatic endocrine progenitor marker *ascl1b* (*Figure 4—figure supplement 1C–J*). Because the ectopic β-cells induced by the *npas4l* mutation also resided in the mesenchymal region, and *npas4l* can act cell-autonomously to

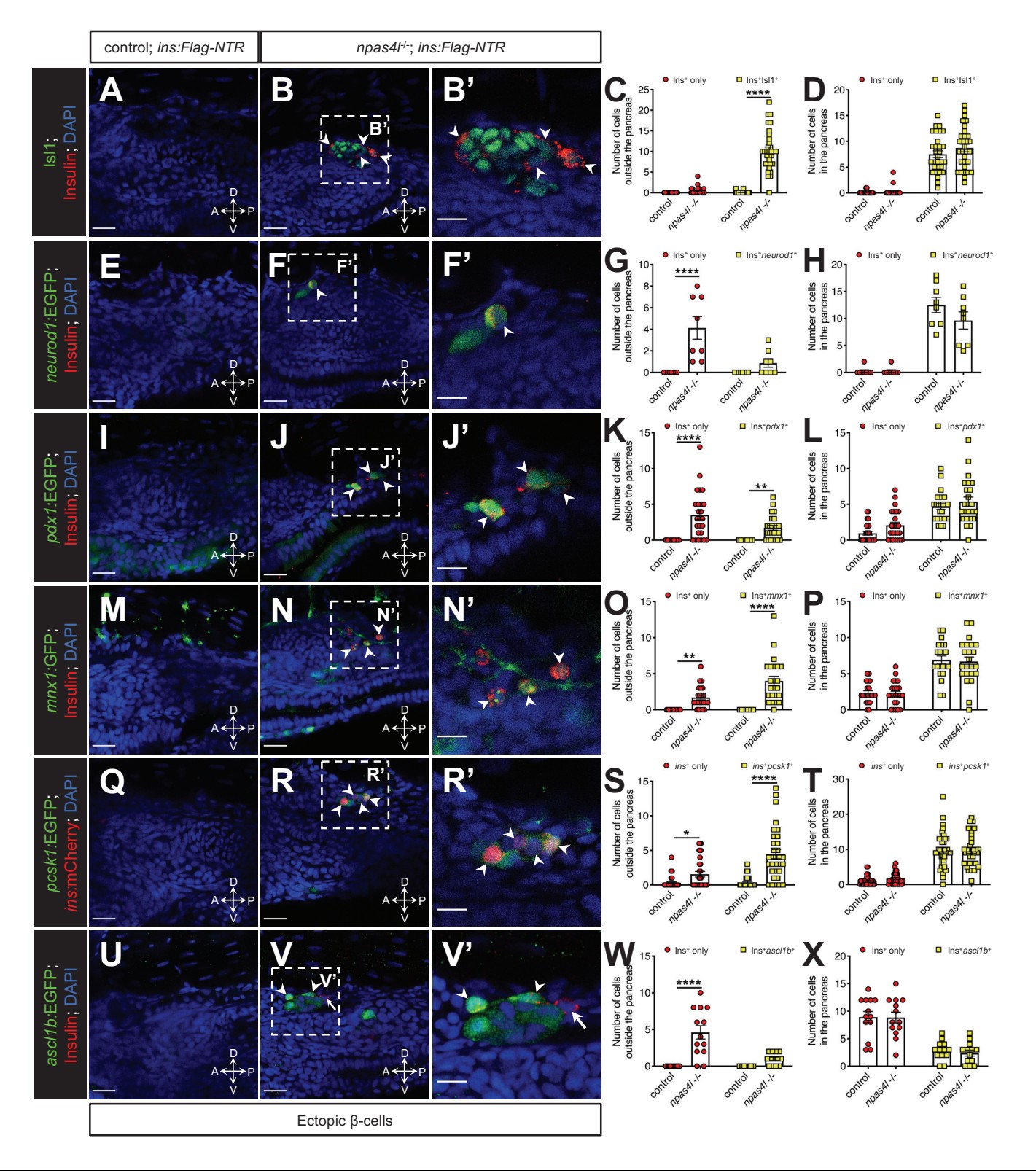

**Figure 2.** The ectopic β-cells co-express insulin and endocrine markers in *npas4l* mutants. Representative confocal images of the tissues adjacent to the pancreas of control siblings and *npas4l*^−/−^ *Tg(ins:Flag-NTR)* zebrafish larvae at 3 dpf after β-cell ablation by MTZ at 1–2 dpf, displaying cells expressing pancreatic endocrine cell markers Isl1 (A–B'), *neurod1* (E–F'), *pdx1* (I–J'), *mnx1* (M–N'), *pcsk1* (Q–R'), and *ascl1b* (U–V') in green, and ectopic β-cells in red with immunostaining for insulin (except Q–R' with the mCherry fluorescence driven by the insulin promoter). Arrowheads point to ectopic β-cells

*Figure 2 continued on next page*

*Figure 2 continued*

that express corresponding markers. Arrows point to β-cells that do not express *ascl1b* (**V and V'**). **B'**, **F'**, **J'**, **N'**, **R'**, and **V'** are magnified from the areas indicated by the white dashed square in **B**, **F**, **J**, **N**, **R**, and **V**, respectively. Quantification of β-cells with or without corresponding marker expression in the ectopic location (**C, G, K, O, S, and W**) or in the pancreas (**D, H, L, P, T, and X**) per larva at 3 dpf. *p=0.0310, **p=0.0039, and ****p<0.0001 (Šidák's multiple comparisons test); (**C, D**) n = 30 (control) and 31 (*npas4l⁻/⁻*); (**G, H**) n = 8 (control) and 8 (*npas4l⁻/⁻*); (**K, L**) n = 25 (control) and 24 (*npas4l⁻/⁻*); (**O, P**) n = 21 (control) and 23 (*npas4l⁻/⁻*); (**S, T**) n = 40 (control) and 32 (*npas4l⁻/⁻*); (**W and X**) n = 13 (control) and 13 (*npas4l⁻/⁻*). Data are represented as the mean ± SEM. Scale bars = 20 μm except **B'**, **F'**, **J'**, **N'**, **R'**, and **V'** (10 μm). Anatomical axes: D (dorsal), V (ventral), A (anterior), and P (posterior). The online version of this article includes the following figure supplement(s) for figure 2:

**Figure supplement 1.** Mutation of *npas4l* inhibits *pdx1*-expressing pancreatic duct formation.

affect the hematopoietic and endothelial lineages (*Parker and Stainier, 1999*), we hypothesised that the ectopic β-cells originated from a mesodermal lineage.

To determine whether the mesoderm was the origin of the ectopic β-cells, we genetically traced the mesodermal cells using *drl:CreER^T2^*, a tamoxifen-inducible Cre transgene driven by a *drl* promoter (*Mosimann et al., 2015*). The spatial expression pattern of *drl* in the *npas4l* mutants resembled that in the sibling controls (*Figure 4—figure supplement 2*), suggesting that *npas4l* mutation did not induce any ectopic expression of *drl* to disrupt the lineage-tracing approach. In combination with a −3.5ubb:LOXP-EGFP-LOXP-mCherry (*ubi:Switch*) reporter (*Mosimann et al., 2011*), the *drl*-expressing mesodermal cells were labelled in red in *Tg(drl:CreER^T2^);Tg(ubi:Switch);Tg(ins:Flag-NTR)* (*drl*-tracing) zebrafish larvae after 4-hydroxytamoxifen (4-OHT) induction (*Figure 4A*, *Figure 4—figure supplement 3*). We treated the transgenic embryos with 4-OHT at 10–12 hpf. We chose to label the mesodermal cells during this period as neither endothelial/hematopoietic cells nor β-cells have developed at that stage, i.e., to exclude confounding effects of endothelial/hematopoietic cells or possible ectopic expression of the lineage tracer in the β-cells of the *npas4l* mutant. To ablate the β-cells, we incubated the 4-OHT-treated transgenic embryos in MTZ at 1–2 days postfertilisation (dpf). We allowed the β-cells to regenerate for 30 hr before we fixed the larvae at 3 dpf for immunostaining (*Figure 4B*).

Immunostaining against insulin displayed a normal set of β-cells in the pancreas of the *drl*-tracing larvae with or without *npas4l* mutation after 30 hr of regeneration (*Figure 4C,E, and F*). In line with the findings shown in *Figure 1*, the *npas4l* mutation induced the formation of ectopic β-cells in the mesenchymal region (*Figure 4D,G, and H*). Furthermore, 98.9% of the ectopic β-cells in the mesenchymal region were mCherry-positive (*Figure 4H'–H'''*), indicating that they derived from the *drl*-expressing mesodermal cells.

With a similar setting, we injected the *drl*-tracing embryos (without any *npas4l* mutation) with control or *etsrp* morpholino at the one-cell stage. Npas4l is essential for the expression of *etsrp*, which is a key regulator of endothelial/myeloid cell specification and vasculogenesis (*Reischauer et al., 2016*; *Sumanas and Lin, 2006*). Similar to the *npas4l* mutation, knocking down *etsrp* led to the formation of ectopic β-cells (*Figure 4I–L'''*). Co-injecting lower doses of two different *etsrp*

**Table 1.** Percentages of pancreatic or ectopic cells co-expressing insulin and corresponding marker gene or protein in *npas4l* mutants.

| Marker gene/protein | Pancreatic co-expression (%) | Ectopic co-expression (%) |
|---|---|---|
| Isl1 | 97.9 | 96.8 |
| *neurod1* | 97.5 | 17.5 |
| *pdx1* | 72.1 | 32.8 |
| *mnx1* | 75.3 | 70.2 |
| *pcsk1* | 85.1 | 74.1 |
| *ascl1b* | 21.2 | 17.8 |

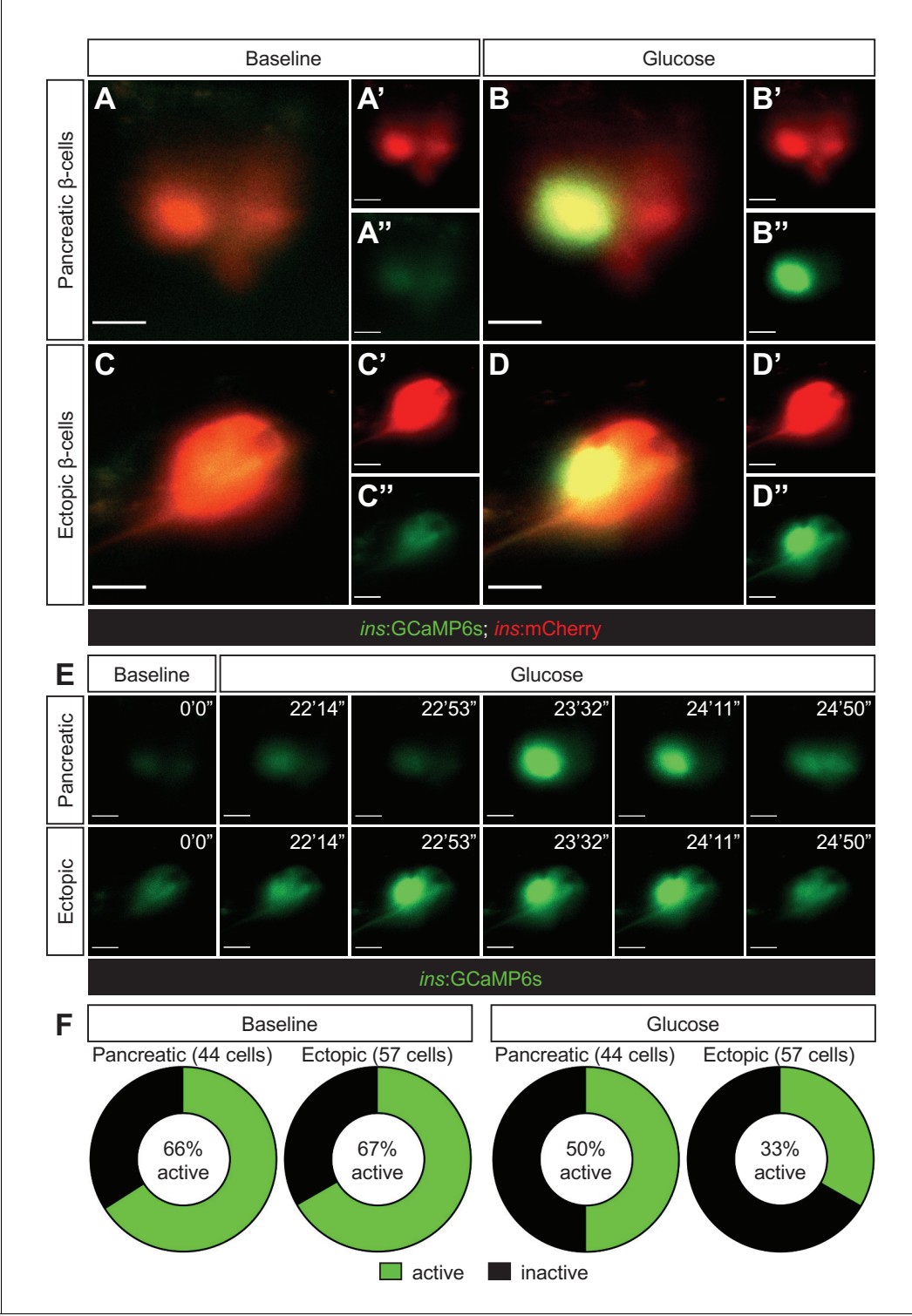

**Figure 3.** Ectopic β-cells can respond to glucose by displaying calcium oscillations. Representative images captured from a live calcium imaging video of *npas4l*$^{-/-}$ *Tg(ins:GCaMP6s);Tg(ins:mCherry);Tg(ins:Flag-NTR)* zebrafish larvae at 3 dpf after β-cell ablation by MTZ from 1 to 2 dpf, showing β-cells expressing *ins:mCherry* in red and displaying calcium activity in green in the pancreatic (A, B) or ectopic (C, D) domain at the baseline (A, C) or after the addition of 200 mM of glucose to the E3 medium (B, D). (E) Sequential images captured from a live calcium imaging video to show the oscillation of calcium signalling at the baseline and after the glucose treatment, displaying the GCaMP6s signal in green. (F) Quantification of pancreatic or ectopic β-cells displaying calcium activity at the baseline or after glucose treatment in 11 (pancreatic) or 17 (ectopic) *npas4l* zebrafish mutants. Scale bars = 10 μm.

morpholinos (MO1 and MO2) exerted a synergistic effect on ectopic β-cell emergence (*Figure 4—figure supplement 4*), suggesting that the phenotype was not due to an off-target effect. The majority of the ectopic β-cells (94.3%) in *etsrp* morphants was also lineage-traced back to the *drl*-expressing mesodermal cells, suggesting that the ectopic β-cell formation was also of mesodermal origin following *etsrp* knockdown.

## The mesodermal cells lose *npas4l* expression after differentiating into ectopic β-cells

We then further examined the autonomous role of the progenitors of endothelial and hematopoietic cells in ectopic β-cell emergence in *npas4l* mutants by tracing the *npas4l* cell lineage with a zebrafish knock-in line, *npas4l*<sup>Pt(+36-npas4l-p2a-Gal4-VP16)bns423</sup>. This knock-in line not only carries a +36 bp mutation in exon 2 of *npas4l* (*npas4l*<sup>bns423</sup> mutation) but also expresses the transcription activator Gal4 under control of the endogenous *npas4l* promoter (*npas4l:Gal4*), allowing us to trace the *npas4l* lineage after crossing it into *Tg(UAS:Cre);Tg(ubi:Switch)* (*Figure 5A*).

We obtained a transheterozygous *npas4l* mutant (*npas4l*<sup>s5/bns423</sup>) by crossing this *npas4l* tracer carrying the *npas4l*<sup>bns423</sup> mutation into the *npas4l*<sup>s5</sup> mutant used in the studies above, and observed similar ectopic β-cell formation (*Figure 5B–E'''*) as found in the homozygous *npas4l*<sup>s5</sup> mutant (*Figure 1*). Some of these ectopic β-cells (32.6%) could be traced back to the *npas4l* lineage as they were labelled by *ubb:mCherry* after the Cre-recombination driven by *npas4l:Gal4* and *UAS:Cre* (*Figure 5C and E–E'''*), which further supports a mesodermal origin of the ectopic β-cells.

By crossing *Tg(npas4l:Gal4)* into *Tg(UAS:EGFP)*, the endogenous promoter activity of *npas4l* could be revealed by EGFP (*Figure 5F*). The ectopic β-cells emerged in close proximity to mesodermal cells with active *npas4l* expression in the transheterozygous *npas4l*<sup>s5/bns423</sup> mutants (*Figure 5G–G'''*). However, we did not observe any ectopic β-cells with EGFP expression, suggesting that these mesodermal cells have lost the identity of the lateral plate mesoderm with *npas4l* expression and started to express insulin as in the endodermal pancreatic β-cells instead.

## The ectopic β-cells derive from the *etsrp*-expressing mesodermal lineage in *etsrp* morphants

To confirm the origin of the ectopic β-cell using a different lineage-tracing approach, we generated *Tg(etsrp:Cre)* zebrafish, based on the promoter of −2.3*estrp:GFP* that has been demonstrated to closely recapitulate the endogenous *etsrp* expression in the lateral plate mesoderm and vasculature (*Veldman and Lin, 2012*). We then crossed *Tg(etsrp:Cre)* into *Tg(ubi:Switch);Tg(ins:Flag-NTR)*, and thereby labelling descendants of the *etsrp* lineage in red (*Figure 6A*). We validated the efficiency of the *etsrp*-lineage tracer and revealed that the majority of *kdrl*-expressing endothelial cells were being traced in the intersegmental vessels (86.6%) and other vasculature (*Figure 6—figure supplement 1*). At the one-cell stage, we injected the *etsrp*-tracing embryos with control or *etsrp* morpholinos. After β-cell ablation by MTZ treatment at 1–2 dpf and β-cell regeneration for 30 hr ectopic β-cells formed in the *etsrp* morphants, and 73.9% of the ectopic β-cells were labelled in red (*Figure 6B–E'''*), illustrating that the *etsrp*-expressing lineage gave rise to a significant portion of the ectopic β-cells.

Moreover, we replaced *ubi:Switch* with *ins:LOXP-mCherry-LOXP-Hsa.HIST1H2BJ-GFP* (*ins:CSH*) in the *etsrp*-tracing zebrafish larvae to

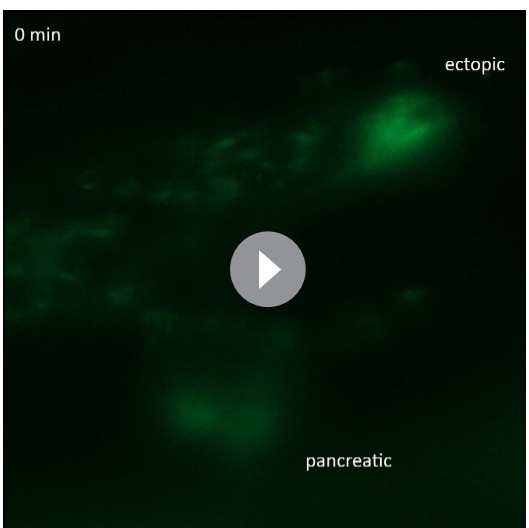

**Video 1.** Live calcium imaging of β-cells in a *npas4l* mutant. An example video of live calcium imaging of both pancreatic and ectopic β-cells in a *Tg(ins:GCaMP6s);Tg(ins:mCherry);Tg(ins:Flag-NTR) npas4l* mutant at 3 dpf after β-cell ablation from 1 to 2 dpf, displaying GCaMP6s signal in green. Glucose (200 mM) was added to the E3 media at around 9 min in the video.
https://elifesciences.org/articles/65758#video1

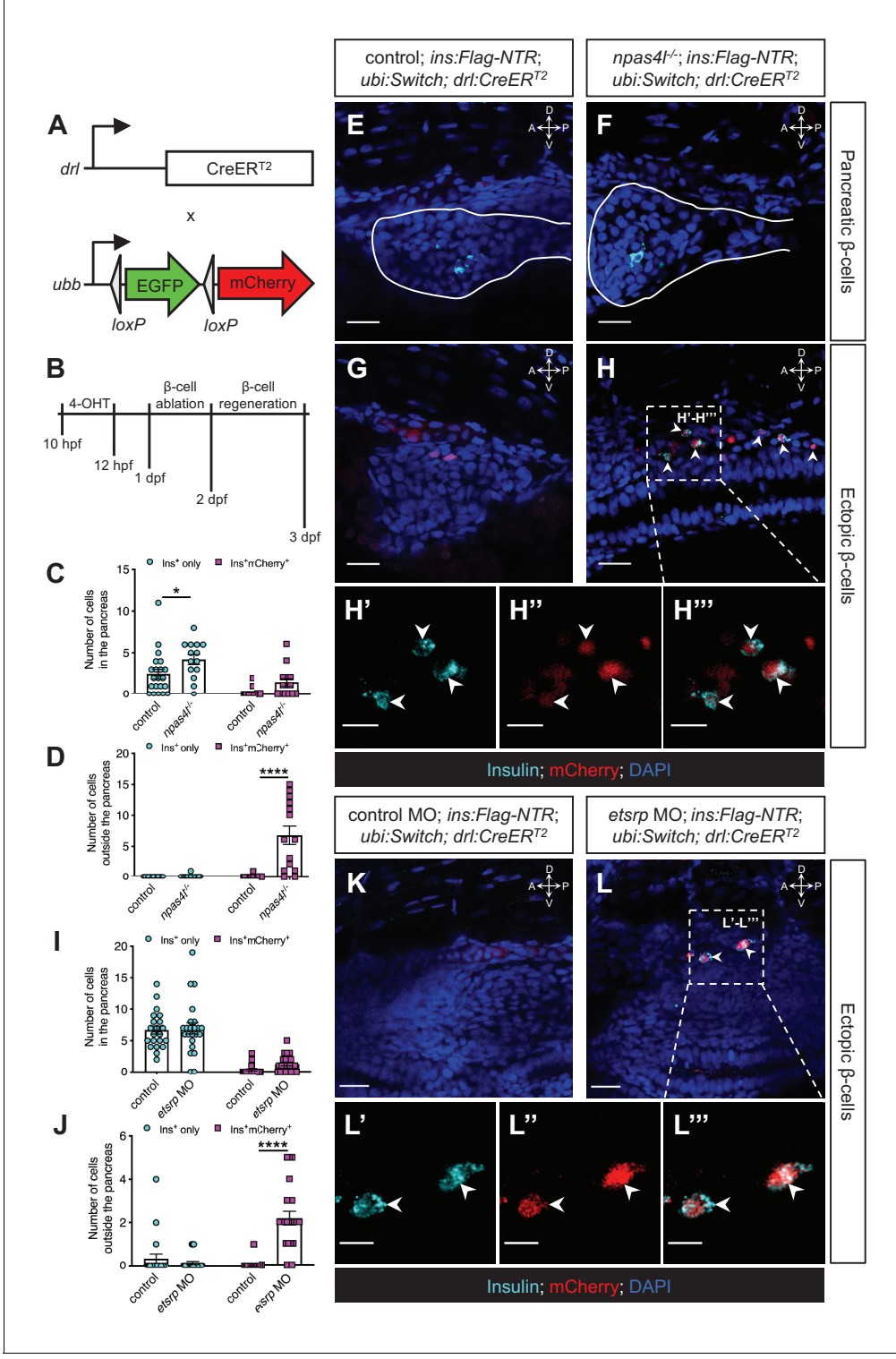

**Figure 4.** The ectopic β-cells are of mesodermal origin in *npas4l* mutants and *etsrp* morphants. (**A**) Constructs of −6.35*drl:Cre-ER^T2* (*drl:CreER^T2*) and −3.5*ubb:LOXP-EGFP-LOXP-mCherry* (*ubi:Switch*). Upon 4-OHT induction between 10 and 12 hpf, Cre recombinase expressed by the *drl* promoter cleave the *loxP* sites to allow *ubb:mCherry* expression in the cells that once expressed *drl*. (**B**) Scheme for tracing the mesodermal lineage of ectopic β-cells in control siblings and *npas4l^−/−* Tg(*ins:Flag-NTR*);Tg(*ubi:Switch*);Tg(*drl:CreER^T2*) zebrafish larvae. (**C, D, I, and J**) Quantification of the pancreatic or ectopic β-cells with or without mesodermal lineage in *npas4l* mutants (**C, D**) or *etsrp* morpholino (MO)-injected larvae (**I, J**) at 3 dpf. *p=0.0227 and ****p<0.0001 (Šidák's multiple comparisons test); (**C, D**) n = 21 (control) and 14 (*npas4l^−/−*) or (**I, J**) n = 21 (control) and 23 (*etsrp* MO). Data are represented as the mean ± SEM. Standard control MO (4 ng) and *etsrp* MO2 (4 ng) were injected to the control group and *etsrp* MO group

*Figure 4 continued on next page*

*Figure 4 continued*

respectively at the one-cell stage (I, J, K–L‴). (E–H‴ and K–L‴) Representative confocal images of pancreatic (E, F) or ectopic β-cells (G, H) of control siblings and *npas4l*⁻/⁻, or ectopic β-cells in control or *etsrp* MO-injected (K, L) *Tg(ins:Flag-NTR);Tg(ubi:Switch);Tg(drl:CreER^T2^)* zebrafish larvae at 3 dpf after β-cell ablation by MTZ at 1–2 dpf, displaying β-cells in cyan with immunostaining for insulin and lineage-traced cells derived from *drl*-expressing mesodermal cells in red from the Cre-recombined *ubi:Switch*. Pancreata are outlined by solid white lines (E, F). Arrowheads point to ectopic β-cells derived from the mesoderm (H–H‴ and L–L‴). Selected areas in dashed squares in (H) and (L) are magnified in split (H′, H″, L′ and L″) and merged (H‴ and L‴) channels, respectively. Scale bars = 20 μm (E–H, K, and L) or 10 μm (H′–H‴ and L′–L″). Anatomical axes: D (dorsal), V (ventral), A (anterior), and P (posterior).

The online version of this article includes the following figure supplement(s) for figure 4:

**Figure supplement 1.** Cell population with reduced *npas4l* expression remains in the lateral plate mesoderm before β-cell ablation.

**Figure supplement 2.** The *npas4l* mutant does not display altered expression of *drl* in the lateral plate mesoderm.

**Figure supplement 3.** Lateral plate mesoderm-derived cardiomyocytes are traced back to a *drl*-expressing lineage.

**Figure supplement 4.** Validation of *etsrp* morpholinos.

directly trace *insulin*-expressing cells originating from the *etsrp*-expressing mesodermal lineage (*Figure 6F*). The co-localisation of insulin staining and the nuclear green tracer further confirms the mesodermal lineage of the ectopic β-cells (*Figure 6G–G‴*).

Together, we used three different lineage-tracing models as well as three different loss of function models, that is using the promoter of *drl*, *npas4l*, or *etsrp* to drive Cre in *npas4l^s5/s5^* mutants, *npas4l^s5/bns423^* mutants, or *etsrp* morphants. This suggests that the ectopic β-cell formation is not restricted to the loss of a specific gene, but rather due to the perturbation of endothelial/myeloid specification.

## Discussion

In this study, we first examined the role of blood vessels in β-cell regeneration in the *cloche* zebrafish mutant, which carries a homozygous *npas4l* mutation (*Reischauer et al., 2016*). We then unexpectedly revealed β-cells regenerating ectopically in the mesenchymal area. The ectopic β-cells were likely functional because they expressed several endocrine and β-cell markers, including Isl1, *mnx1*, and *pcsk1*, and were capable of responding to glucose to induce calcium oscillations during β-cell regeneration, although we do not know if they possess all the features of bona fide β-cells. By combining in situ hybridisation, lineage tracing, and confocal microscopy, we successfully traced the origin of the ectopic β-cells to the mesodermal lineage. A recent study has reported the conversion of Etsrp-deficient vascular progenitors into skeletal muscle cells and highlighted the plasticity of mesodermal cell fate determination within the same germ layer (*Chestnut et al., 2020*). Our data demonstrated the plasticity of β-cell differentiation across the committed germ layers in vivo, i.e., switching from a mesodermal to an endodermal fate in a regenerative setting, while gastrulation and cell fate commitment in the germ layers are considered to be irreversible in development. Ectopic pancreata have been observed before, e.g., in *Hes1* mutant mice (*Fukuda, 2006*; *Sumazaki et al., 2004*), although that has been shown to be through an expansion of the pancreas rather than through changes in cell fate determination across organs or germ layers (*Jørgensen et al., 2018*). Our discovery is, to our knowledge, the first demonstration of ectopic β-cells with a mesodermal origin in vivo.

The recent genome-wide study with zebrafish embryos has confirmed the crucial role of *npas4l* in the early specification of endothelium and blood as the expressions of some endothelial and hematopoietic genes like *tal1*, *etsrp*, *lmo2*, *gfi1aa*, and *gata1a* were downregulated in homozygous *npas4l* mutants (*Marass et al., 2019*), suggesting that some populations of mesodermal cells may not have acquired their designated cell fates and remained in a more plastic state. In line with the important role of *etsrp* in the endothelial cell fate determination, *Chestnut et al., 2020* have reported a significant reduction in the cell clusters of endothelial cells and endothelial progenitor cells in homozygous *etsrp* mutants in a single-cell RNA-seq analysis. Interestingly, they have also shown a remarkable increase in the cluster of lateral plate mesoderm, which points to the possibility

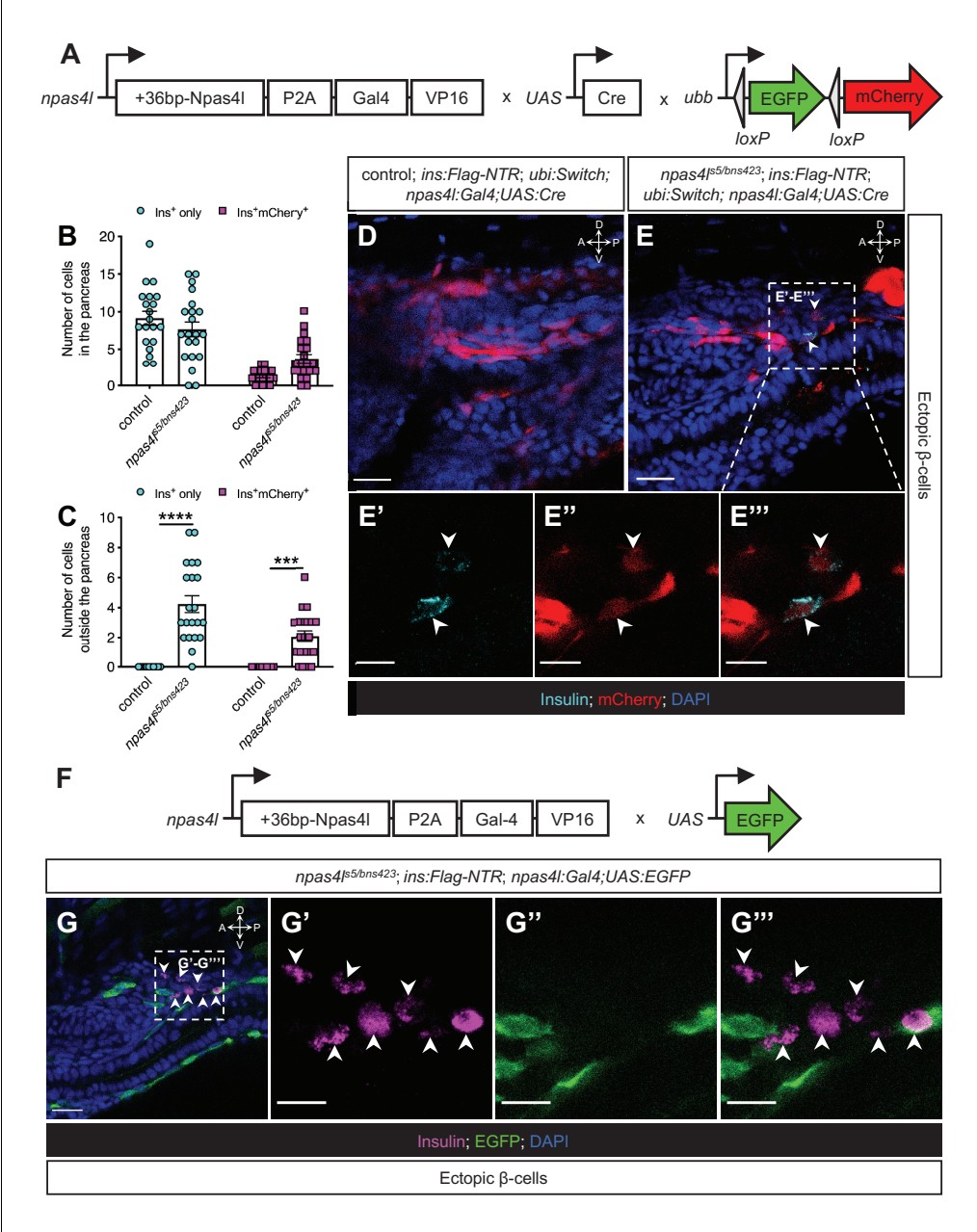

**Figure 5.** The mesodermal cells lose *npas4l* expression after differentiating into ectopic β-cells. (**A**) Schematics of *npas4l*[Pt(+36-npas4l-p2a-Gal4-VP16)bns423] (*npas4l:Gal4*), *UAS:Cre* and −3.5ubb:LOXP-EGFP-LOXP-mCherry (*ubi:Switch*). (**B, C**) Quantification of the pancreatic or ectopic β-cells with or without *npas4l*-positive mesodermal origin in control or transheterozygous *npas4l*[s5/bns423] mutants at 3 dpf. ***p=0.0001 and ****P<0.0001 (Šidák's multiple comparisons test); n = 20 (control) and 21 (*npas4l*[s5/bns423]). Data are represented as the mean ± SEM. (**D–E'''**) Representative confocal images of ectopic β-cells and *npas4l*-positive lineage-traced cells in control siblings and *npas4l*[s5/bns423] *Tg(ins:Flag-NTR);Tg(ubi:Switch);Tg(npas4l:Gal4);Tg(UAS:Cre)* zebrafish larvae at 3 dpf after β-cell ablation by MTZ from 1 to 2 dpf, displaying β-cells in cyan with immunostaining for insulin and lineage-traced cells derived from *npas4l*-expressing mesodermal cells in red from the Cre-recombined *ubi:Switch*. The selected area in a dashed square in (**E**) is magnified in split (**E'** and **E''**) and merged (**E'''**) channels, respectively. Arrowheads point to ectopic β-cells derived from the mesoderm (**E–E'''**). (**F**) Schematics of *npas4l*[Pt(+36-npas4l-p2a-Gal4-VP16)bns423] (*npas4l:Gal4*) and *UAS:EGFP*. (**G–G'''**) Representative confocal images of ectopic β-cells losing *npas4l* expression in *npas4l*[s5/bns423] *Tg(ins:Flag-NTR); Tg(npas4l:Gal4);Tg(UAS:EGFP)* zebrafish larvae at 3 dpf after β-cell ablation by MTZ from 1 to 2 dpf, displaying β-cells in magenta with immunostaining for insulin and cells expressing *npas4l* in green. The selected area in a dashed square in (**G**) is magnified in split (**G'** and **G''**) and merged (**G'''**) channels, respectively. Arrowheads point to ectopic β-cells without *npas4l* expression (**G–G'''**). Scale bars = 20 μm (**D, E, and G**) or 10 μm (**E'–E'''** and **G'–G''**). Anatomical axes: D (dorsal), V (ventral), A (anterior), and P (posterior).

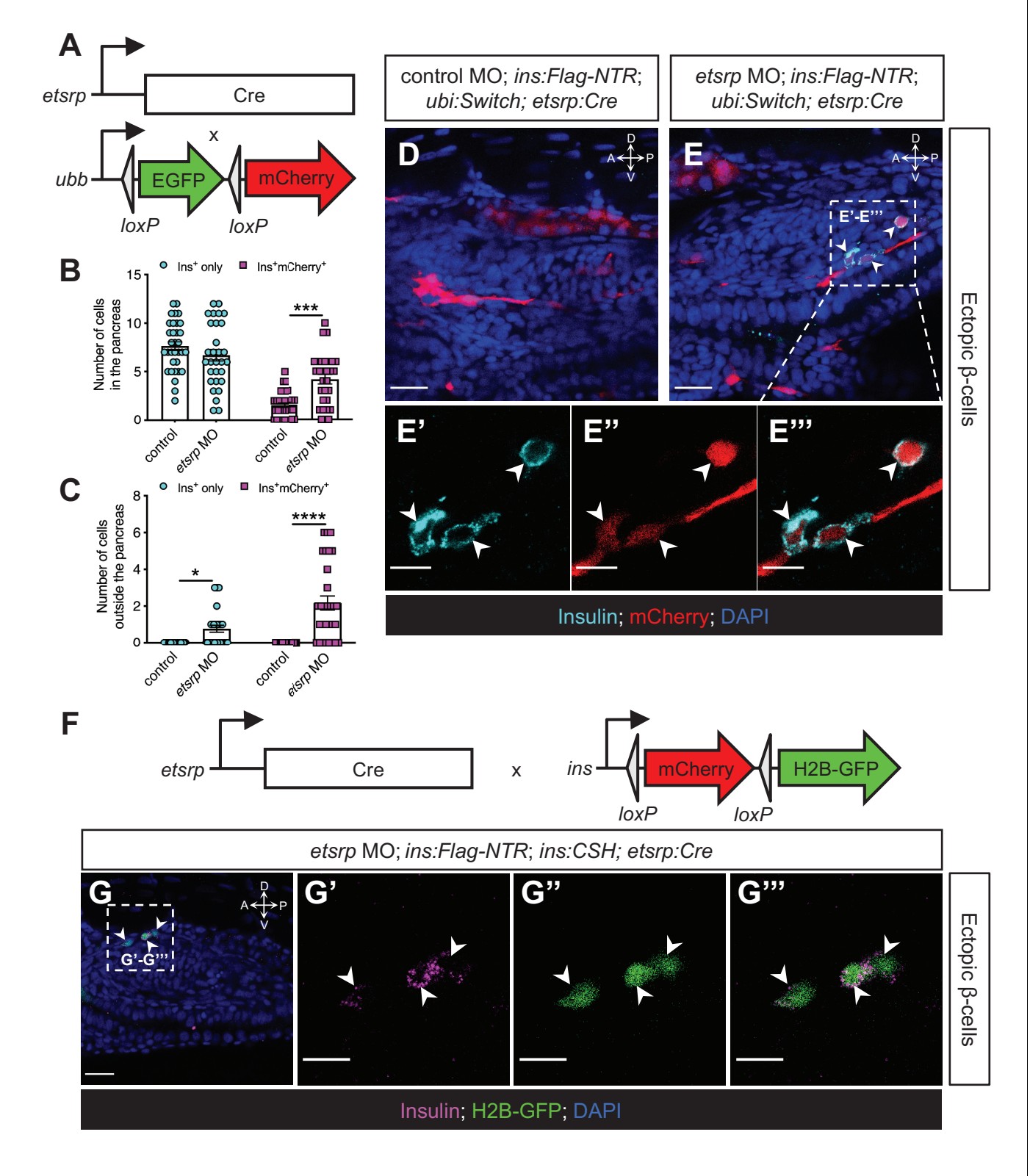

**Figure 6.** The ectopic β-cells derive from the *etsrp*-expressing mesodermal lineage in *etsrp* morphants. (**A**) Constructs of *etsrp:Cre* and −3.5ubb:LOXP-EGFP-LOXP-mCherry (*ubi:Switch*). (**B, C**) Quantification of the pancreatic or ectopic β-cells with or without *etsrp*-positive mesodermal origin in control or *etsrp* morpholino (MO)-injected larvae at 3 dpf. *p=0.0169, ***p=0.0002, and ****p<0.0001 (Šidák's multiple comparisons test); n=33 (control) and 30 (*etsrp* MO). Data are represented as the mean ± SEM. (**D–E′′**) Representative confocal images of ectopic β-cells and *etsrp*-positive lineage-traced cells

*Figure 6 continued on next page*

*Figure 6 continued*

in control or *etsrp* MO-injected *Tg(ins:Flag-NTR);Tg(ubi:Switch);Tg(etsrp:Cre)* zebrafish larvae at 3 dpf after β-cell ablation by MTZ at 1–2 dpf, displaying β-cells in cyan with immunostaining for insulin and lineage-traced cells derived from *etsrp*-expressing mesodermal cells in red from the Cre-recombined *ubi:Switch*. The selected area in a dashed square in (E) is magnified in split (E', E'') and merged (E''') channels, respectively. Arrowheads point to ectopic β-cells derived from the mesoderm (E–E'''). (F) Constructs of *etsrp:Cre* and *ins:LOXP-mCherry-LOXP-Hsa.HIST1H2BJ-GFP* (*ins:CSH*). (G–G''') Representative confocal images of ectopic β-cells derived from the *etsrp*-expressing lineage in *etsrp* MO-injected *Tg(ins:Flag-NTR);Tg(ins:CSH);Tg (etsrp:Cre)* zebrafish larvae at 3 dpf after β-cell ablation by MTZ at 1–2 dpf, displaying β-cells in magenta with immunostaining for insulin and lineage-traced cells derived from *etsrp*-expressing mesodermal cells in nuclear green from the Cre-recombined *ins:CSH*. The selected area in a dashed square in (G) is magnified in split (G' and G'') and merged (G''') channels, respectively. Scale bars = 20 μm (D, E, and G) or 10 μm (E'–E''' and G'–G'''). Anatomical axes: D (dorsal), V (ventral), A (anterior), and P (posterior).

The online version of this article includes the following figure supplement(s) for figure 6:

**Figure supplement 1.** Most of the *kdrl*-expressing endothelial cells are traced back to an *etsrp* lineage.

that a larger pool of more versatile mesodermal cells remained in the homozygous *etsrp* mutants. However, it is difficult to deduce why those mesodermal cells with perturbed cell fates would be competent to differentiate into ectopic insulin-expressing β-cells from these transcriptomic data; because the population of these competent mesodermal cells might be very small according to the number of ectopic β-cells that we observed and their single-cell RNA-seq was not performed after β-cell ablation.

Since we could trace the origin of ectopic β-cells back to *drl*, *npas4l*, and *etsrp* lineages, we believe that the versatile lateral plate mesoderm close to the pancreas could contribute to the β-cell regeneration in *npas4l* mutants or *etsrp* morphants. Tracing the lineages with early mesodermal, endodermal, and endocrine markers, and sorting these lineage-traced cells could be necessary for a single-cell transcriptomic study to fully understand the cellular status of this small population of versatile cells and further elucidate the underlying molecular mechanisms. Before deciphering these

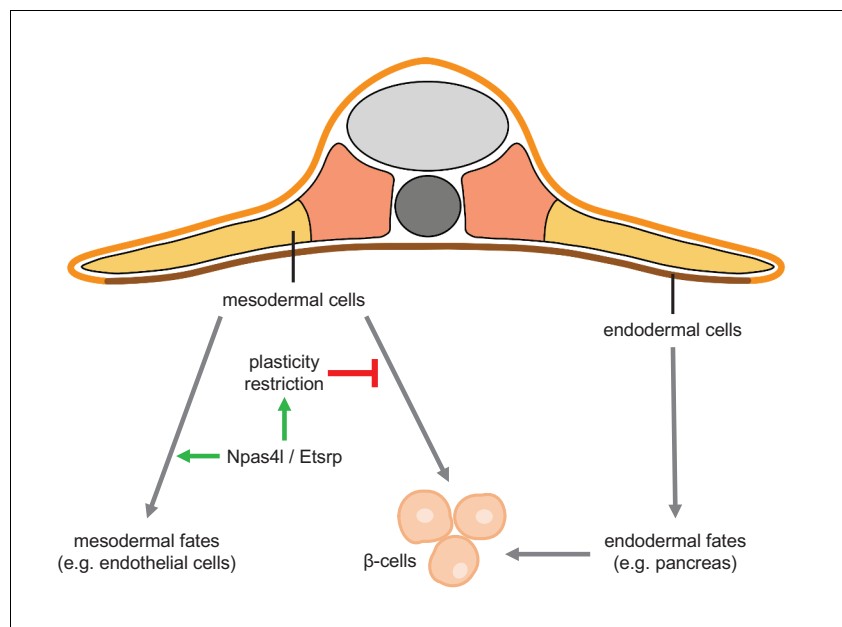

**Figure 7.** Npas4l/Etsrp restricts the plasticity of the mesoderm. Mesoderm and endoderm normally follow Waddington's landscape model to further differentiate into cells with mesodermal fates and endodermal fates, respectively, during development. However, mutating *npas4l* or knocking down *etsrp* not only abolishes the endothelial/myeloid specification but also induces plasticity of mesodermal cells to enable their differentiation into β-cells across the germ layer boundary.

mechanisms, we cannot formally rule out the possibility that the mechanisms of ectopic β-cell formation in *npas4l* mutants and *etsrp* morphants could be different.

The mutated gene in the *cloche* mutant was named *npas4l* because its encoded protein shares homology with human NPAS4 (*Reischauer et al., 2016*). Although injecting human *NPAS4* mRNA or zebrafish *npas4l* mRNA into zebrafish *cloche* mutant embryos at the one-cell stage could rescue the mutants, *Npas4* knockout mice are unlikely to share the same severe vascular and hematopoietic defects as zebrafish *npas4l* mutants because *Npas4* knockout mice survive to adulthood (*Lin et al., 2008*). This discrepancy suggests that other members of the mammalian NPAS protein family or other proteins may be functionally redundant with NPAS4 in vascular and hematopoietic development. Mammalian NPAS4 has been shown to have important cell-autonomous functions in β-cells (*Sabatini et al., 2018*; *Speckmann et al., 2016*). In zebrafish, *npas4a* is the main *npas4* paralogue expressed in β-cells (*Tarifeño-Saldivia et al., 2017*), meaning that it is unlikely the phenotype we identified early in development in *npas4l* mutants is related to the functions of Npas4 in β-cells. Further studies on NPAS4, related bHLH transcription factors, and ETV2 in mammals should elucidate whether inactivating such factors promotes β-cell formation with or without significantly perturbing the development of blood cells and vessels.

In summary, we have shown that the *npas4l* mutation and *etsrp* knockdown each induces ectopic regeneration of β-cells from the mesoderm. Our findings suggest a plasticity potential of mesodermal cells to differentiate into endodermal cells including β-cells (*Figure 7*). Future studies on the restriction of this plasticity would not only increase our understanding of the gating role of Npas4l and Etsrp in cell fate determination but also help to exploit an alternative source for β-cell regeneration.

# Materials and methods

## Key resources table

| Reagent type (species) or resource | Designation | Source or reference | Identifiers | Additional information |
|---|---|---|---|---|
| Genetic reagent (*Danio rerio*) | *npas4l*$^{s5}$ | PMID:7588049 | ZFIN: ZDB-ALT-010426–6 | |
| Genetic reagent (*Danio rerio*) | *npas4l*$^{Pt(+36-npas4l-p2a-Gal4-VP16)bns423}$ | Other | | Will be described in detail elsewhere (K.M. and D.Y.R.S., Manuscript in preparation) |
| Genetic reagent (*Danio rerio*) | *Tg(ins:Hsa.HIST1H2BJ-GFP;ins:DsRed)*$^{s960}$ | PMID:25117518 | ZFIN: ZDB-ALT-131001–6 | |
| Genetic reagent (*Danio rerio*) | *Tg(ins:FLAG-NTR, cryaa:mCherry)*$^{s950}$ | PMID:22608007 | ZFIN: ZDB-ALT-130930–5 | |
| Genetic reagent (*Danio rerio*) | *Tg(ptf1a:EGFP)*$^{jh1}$ | PMID:16258076 | ZFIN: ZDB-ALT-070531–2 | |
| Genetic reagent (*Danio rerio*) | *Tg*$^{BAC}$*(hand2:EGFP)*$^{pd24}$ | PMID:21397850 | ZFIN: ZDB-ALT-110128–40 | |
| Genetic reagent (*Danio rerio*) | *Tg*$^{BAC}$*(neurod1:EGFP)*$^{nl1}$ | PMID:18305245 | ZFIN: ZDB-ALT-080701–1 | |
| Genetic reagent (*Danio rerio*) | *Tg*$^{BAC}$*(pdx1:EGFP)*$^{bns13}$ | PMID:31142539 | ZFIN: ZDB-ALT-191212–6 | |
| Genetic reagent (*Danio rerio*) | *Tg(mnx1:GFP)*$^{ml2}$ | PMID:16162647 | ZFIN: ZDB-ALT-051025–4 | |
| Genetic reagent (*Danio rerio*) | *Tg(pcsk1:EGFP)*$^{Kl106}$ | PMID:27516442 | ZFIN: ZDB-ALT-161115–10 | |
| Genetic reagent (*Danio rerio*) | *Tg*$^{BAC}$*(ascl1b:EGFP-2A-Cre-ERT2)*$^{ulg006}$ | PMID:26329351 | ZFIN: ZDB-ALT-160205–2 | |
| Genetic reagent (*Danio rerio*) | *Tg(ins:GCaMP6s, cryaa:RFP)* | PMID:28939870 | ZFIN: ZDB-ALT-181221–2 | |

*Continued on next page*

*Continued*

| Reagent type (species) or resource | Designation | Source or reference | Identifiers | Additional information |
|---|---|---|---|---|
| Genetic reagent (*Danio rerio*) | *Tg(−3.5ubb:LOXP-EGFP-LOXP-mCherry)*^cz1701 | PMID:21138979 | ZFIN: ZDB-ALT-110124–1 | |
| Genetic reagent (*Danio rerio*) | *Tg(−6.35drl:Cre-ERT2,cryaa:Venus)*^cz3333 | PMID:26306682 | ZFIN: ZDB-ALT-160129–4 | |
| Genetic reagent (*Danio rerio*) | *Tg(5xUAS:EGFP)*^nkuasgfp1a | PMID:18202183 | ZFIN: ZDB-ALT-080528–1 | |
| Genetic reagent (*Danio rerio*) | *Tg(kdrl:EGFP)*^s843 | PMID:16251212 | ZFIN: ZDB-ALT-050916–14 | |
| Genetic reagent (*Danio rerio*) | *Tg(ubb:LOXP-CFP-LOXP-zgc:114046-mCherry)*^jh63 | PMID:25773748 | ZFIN: ZDB-ALT-151007–31 | |
| Genetic reagent (*Danio rerio*) | *Tg(ins:LOXP-mCherry-LOXP-Hsa.HIST1H2BJ-GFP,cryaa:Cerulean)*^s934 | PMID:21497092 | ZFIN: ZDB-ALT-111031–2 | |
| Genetic reagent (*Danio rerio*) | *Tg(etsrp:iCre;cryaa:Venus)*^KI114 | This paper | | See Materials and methods, section *Zebrafish* |
| Genetic reagent (*Danio rerio*) | *Tg(UAS:Cre, cryaa:Cerulean)*^bns382 | This paper | | See Materials and methods, section *Zebrafish* |
| Antibody | Anti-GFP (chicken polyclonal) | Aves Labs | GFP-1020 | (1:500) |
| Antibody | Anti-RFP (rabbit polyclonal) | Abcam | ab62341 | (1:500) |
| Antibody | Anti-tdTomato (goat polyclonal) | MyBioSource | MBS448092 | (1:500) |
| Antibody | Anti-insulin (rabbit polyclonal) | Cambridge Research Biochemicals | Customised | (1:100) |
| Antibody | Anti-pan-cadherin (rabbit polyclonal) | Sigma-Aldrich | C3678 | (1:5000) |
| Antibody | Anti-islet-1-homeobox (mouse monoclonal) | DSHB | 40.3A4 supernatant | (1:10) |
| Recombinant DNA reagent | p5E-MCS (plasmid) | Tol2kit | 228 | |
| Recombinant DNA reagent | p5E-*etsrp* (plasmid) | This paper | | −2.3*etsrp* promoter inserted into p5E-MCS |
| Recombinant DNA reagent | pME-iCre (plasmid) | Other | | From Kristen M. Kwan's lab |
| Recombinant DNA reagent | p3E-polyA (plasmid) | Tol2kit | 302 | |
| Recombinant DNA reagent | pDestTol2gY (plasmid) | Other | | From Naoki Tsuji |
| Recombinant DNA reagent | −2.3*etsrp:iCre, cryaa:Venus* (plasmid) | This paper | | See Materials and methods, section *Zebrafish* |
| Sequence-based reagent | −2.3*etsrp*_FWD | This paper | PCR primer | TATAGGGCGAATT GggtaccTTCAGTAA GCAGACTCCTTCAATCA |
| Sequence-based reagent | −2.3*etsrp*_REV | This paper | PCR primer | AGCTGGAGCTC CAccgcggTTCGGCA TACTGCTGTTGGAC |
| Sequence-based reagent | Standard control morpholino | Gene Tools | Morpholino | CCTCTTACCTCA GTTACAATTTATA |

*Continued on next page*

*Continued*

| Reagent type (species) or resource | Designation | Source or reference | Identifiers | Additional information |
|---|---|---|---|---|
| Sequence-based reagent | Random control morpholino | Gene Tools | Morpholino | Mixture of many oligo sequences |
| Sequence-based reagent | *etsrp* morpholino MO1 | Gene Tools | Morpholino ZFIN: ZDB-MRPHLNO-060407–2 | TTGGTACATTTCCA TATCTTAAAGT |
| Sequence-based reagent | *etsrp* morpholino MO2 | Gene Tools | Morpholino ZFIN: ZDB-MRPHLNO-060407–3 | CACTGAGTCCTTAT TTCACTATATC |
| Sequence-based reagent | *npas4l*_ISH_FWD | This paper | PCR primer | ACTCGGGCAT CAGGAGGATC |
| Sequence-based reagent | *npas4l*_ISH_REV | This paper | PCR primer | (CCTAATACGACT CACTATAGGG)GAC ACCAGCATACGA CACACAAC |
| Sequence-based reagent | *drl*_ISH_FWD | This paper | PCR primer | ATGAAGAATACA ACAAAACCC |
| Sequence-based reagent | *drl*_ISH_REV | This paper | PCR primer | (CCTAATACGAC TCACTATAGGG)TGA GAAGCTCTGGCCGC |
| Sequence-based reagent | *npas4ls5*_FWD | PMID:27411634 | PCR primer | TTCCATCTTCT GAATCCTCCA |
| Sequence-based reagent | *npas4ls5*_REV | PMID:27411634 | PCR primer | GGACAGACCC AGATACTCGT |
| Sequence-based reagent | *npas4ls5*_SEQ | This paper | Sequencing primer | TTTCTGCCGTG AATGGATGTG |
| Commercial assay or kit | Gateway LR Clonase II Enzyme Mix | Invitrogen | 11791–020 | |
| Commercial assay or kit | Phusion High-Fidelity DNA Polymerase | Thermo Scientific | F-530L | |
| Commercial assay or kit | In-Fusion HD Cloning Kit | Takara Bio | 639648 | |
| Commercial assay or kit | MAXIscript T7/T3 Transcription Kit | Invitrogen | AM1324 | |
| Chemical compound, drug | Metronidazole | Sigma-Aldrich | M3761 | |
| Chemical compound, drug | 4-Hydroxytamoxifen | Sigma-Aldrich | H7904 | |
| Software, algorithm | ImageJ | PMID:22930834 | | |
| Software, algorithm | Fiji | PMID:22743772 | | |
| Software, algorithm | LAS X Version 3.5.5.19976 | Leica | | |
| Software, algorithm | ZEN 3.1 | Zeiss | | |
| Software, algorithm | GraphPad Prism 9 | GraphPad Software | | |
| Other | DAPI | ThermoFisher Scientific | D1306 | (1 µg/mL) |

## Zebrafish

The following previously published mutant and transgenic zebrafish (*Danio rerio*) lines were used: *cloche*[S5] (*Field et al., 2003*) as the *npas4l*[s5] mutant, *Tg(ins:Hsa.HIST1H2BJ-GFP;ins:DsRed)*[s960] (*Tsuji et al., 2014*) abbreviated as *Tg(ins:H2BGFP;ins:DsRed)*, *Tg(ins:FLAG-NTR,cryaa:mCherry)*[s950] (*Andersson et al., 2012*) abbreviated as *Tg(ins:Flag-NTR)*, *Tg(ptf1a:EGFP)*[jh1] (*Godinho et al., 2005*), *Tg*[BAC]*(hand2:EGFP)*[pd24] (*Kikuchi et al., 2011*), *Tg*[BAC]*(neurod1:EGFP)*[nl1] (*Obholzer et al., 2008*), *Tg*[BAC]*(pdx1:EGFP)*[bns13] (*Helker et al., 2019*), *Tg(mnx1:GFP)*[ml2] (*Flanagan-Steet et al., 2005*), *Tg(pcsk1:EGFP)*[KI106] (*Lu et al., 2016*), *Tg*[BAC]*(ascl1b:EGFP-2A-Cre-ERT2)*[ulg006] (*Ghaye et al., 2015*) abbreviated as *Tg(ascl1b:EGFP)*, *Tg(ins:GCaMP6s,cryaa:RFP)* (*Singh et al., 2017*) generated by reinjecting the plasmid and abbreviated as *Tg(ins:GCaMP6s)*, *Tg(−3.5ubb:LOXP-EGFP-LOXP-mCherry)*[cz1701] (*Mosimann et al., 2011*) abbreviated as *Tg(ubi:Switch)*, *Tg(−6.35drl:Cre-ERT2,cryaa:Venus)*[cz3333] (*Mosimann et al., 2015*) abbreviated as *Tg(drl:CreER*[T2]*)*, *Tg(5xUAS:EGFP)*[nkuasgfp1a] (*Asakawa et al., 2008*) abbreviated as *Tg(UAS:EGFP)*, *Tg(kdrl:EGFP)*[s843] (*Jin et al., 2005*), *Tg(ubb:LOXP-CFP-LOXP-zgc:114046-mCherry)*[jh63] (*Wang et al., 2015*), and *Tg(ins:LOXP-mCherry-LOXP-Hsa.HIST1H2BJ-GFP,cryaa:Cerulean)*[s934] (*Hesselson et al., 2011*), which is referred to *Tg(ins:mCherry)* in *Figures 2* and *3* and *Video 1* and *Tg(ins:CSH)* in *Figure 6*.

The *npas4l*[Pt(+36-npas4l-p2a-Gal4-VP16)bns423] line (abbreviated as *npas4l:Gal4* or *npas4l*[bns423]) was generated using the GeneWeld system (*Wierson et al., 2019*) and will be described in detail elsewhere (K.M. and D.Y.R.S., Manuscript in preparation).

The *Tg(etsrp:iCre;cryaa:Venus)*[KI114] line, abbreviated as *Tg(etsrp:Cre)*, was generated by the Tol2 transposon system, and the construct was made by MultiSite Gateway Cloning (Invitrogen). The amplicon of the −2.3*etsrp* promoter was synthesised from zebrafish genomic DNA with a forward primer 5'-TATAGGGCGAATTGggtaccTTCAGTAAGCAGACTCCTTCAATCA-3' and a reverse primer 5'-AGCTGGAGCTCCAccgcggTTCGGCATACTGCTGTTGGAC-3' by Phusion High-Fidelity DNA Polymerase (Thermo Scientific) as an insert for In-Fusion Cloning (Takara Bio) with p5E-MCS using restriction sites KpnI and SacII to yield p5E-*etsrp*. Subsequently, p5E-*etsrp*, pME-iCre, p3E-polyA, and pDestTol2gY were used for the LR reaction to generate the construct −2.3*etsrp:iCre,cryaa:Venus*. Expressing *iCre* (improved Cre recombinase) by an *etsrp* promoter would cleave the *loxP* sites of reporters, e.g. *ubi:Switch*, which allows us to trace the descendants differentiated from an *etsrp* lineage.

The *Tg(UAS:Cre, cryaa:Cerulean)*[bns382] line, abbreviated as *Tg(UAS:Cre)*, was generated via Tol2-mediated transgenesis by injecting a construct that placed the Cre recombinase coding sequence downstream of five tandem copies of the upstream activation sequence (UAS). The eye-marker *cryaa:Cerulean* cassette (*Hesselson et al., 2009*) was inserted downstream of Cre in the reverse orientation using 0.56 kb of the *cryaa* promoter (*Kurita et al., 2003*).

Males and females ranging in age from 3 months to 2 years were used for breeding to obtain new offspring for experiments. Individuals were sorted into the control sibling group (*npas4l*[+/+], *npas4l*[s5/+], or *npas4l*[bns423/+]) and the *npas4l* mutant group (*npas4l*[s5/s5] or *npas4l*[s5/bns423]) based on the characteristic pericardial oedema and blood-cell deficit. Zebrafish larvae were allocated into different experimental groups based on their phenotypes and genotypes in experiments involving *cloche* mutants. In morpholino knockdown experiments, zebrafish embryos were randomly assigned to each experimental condition for injection. Experimental procedures were performed on the zebrafish from 10 hpf to 3 dpf before the completion of sex determination and gonad differentiation. All zebrafish, except *npas4l* mutants and *etsrp* morphants, appeared healthy and survived to adulthood. The *npas4l* mutants exhibited pericardial oedema, bell-shaped hearts, and blood deficits, as previously reported (*Stainier et al., 1995*). The *etsrp* morphants had similar phenotypes. All studies involving zebrafish were performed in accordance with local guidelines and regulations and approved by regional authorities.

## Chemical ablation of β-cells

As in our previous report (*Schulz et al., 2016*), we ablated β-cells by incubating the β-cell ablation model *Tg(ins:Flag-NTR)* zebrafish in E3 medium supplemented with 10 mM metronidazole (MTZ, Sigma-Aldrich), 1% DMSO (VWR), and 0.2 mM 1-phenyl-2-thiourea (Acros Organics) for 24 hr from 1 to 2 dpf.

## Microinjection of morpholinos

Standard control morpholino (5′-CCTCTTACCTCAGTTACAATTTATA-3′), random control morpholino, *etsrp* morpholino MO1 (5′-TTGGTACATTTCCATATCTTAAAGT-3′), and MO2 (5′-CACTGAG TCCTTATTTCACTATATC-3′) (*Sumanas and Lin, 2006*) were purchased from Gene Tools, LLC. Morpholinos were injected into the one-cell-stage zebrafish embryos at the doses specified in the figure legends.

## Lineage tracing by tamoxifen-inducible cre recombinase

To genetically trace the mesodermal lineage, we treated *Tg(ins:Flag-NTR);Tg(ubi:Switch);Tg(drl: CreER^{T2}^)* zebrafish embryos with 10 μM 4-OHT (Sigma-Aldrich) in E3 medium in 90 mm Petri dishes, with approximately 60 individuals per dish, from 10 to 12 hpf. Upon induction by 4-OHT, cytoplasmic CreER^{T2}^ would be translocated to the nucleus to excise the loxP-flanked EGFP to enable mCherry expression in *drl*-expressing cells and their descendants, indicating a mesodermal lineage.

## Sample fixation for immunostaining

Before fixing the zebrafish larvae, we confirmed the presence of the transgenes by determining the corresponding fluorescent markers and subsequently examined them under a widefield fluorescence microscope LEICA M165 FC (Leica Microsystems). We then euthanised the zebrafish larvae with 250 mg/L tricaine (Sigma-Aldrich) in E3 medium followed by washing in distilled water three times. We fixed the samples in 4% formaldehyde (Sigma-Aldrich) in PBS (ThermoFisher Scientific) at 4°C overnight. After washing away the fixative with PBS three times, we removed the skin and crystallised yolk of the zebrafish larvae by forceps under the microscope to expose the pancreas and mesenchyme for immunostaining.

## Immunostaining and confocal imaging

As in our previous report (*Liu et al., 2018*), we started immunostaining by incubating the zebrafish samples in blocking solution (0.3% Triton X-100, 4% BSA and 0.02% sodium azide from Sigma-Aldrich in PBS) at room temperature for 1 hr. We then incubated the samples in blocking solution with primary antibodies at 4°C overnight. After removing the primary antibodies, we washed the samples with washing buffer (0.3% Triton X-100 in PBS) eight times at room temperature for 2 hr. Afterwards, we incubated the samples in blocking solution with fluorescent dye-conjugated secondary antibodies and the nuclear counterstain DAPI (ThermoFisher Scientific) if applicable at 4°C overnight. Next, we removed the secondary antibodies and nuclear counterstain and washed the samples with washing buffer eight times at room temperature for 2 hr. The following primary antibodies were used: anti-GFP (1:500, Aves Labs, GFP-1020), anti-RFP (1:500, Abcam, ab62341), anti-tdTomato (1:500, MyBioSource, MBS448092), anti-insulin (1:100, Cambridge Research Biochemicals, customised), anti-pan-cadherin (1:5000, Sigma, C3678), and anti-islet-1-homeobox (1:10, DSHB, 40.3A4 supernatant).

Before confocal imaging, we mounted the stained samples in VECTASHIELD Antifade Mounting Medium (Vector Laboratories) on microscope slides with the pancreas facing the cover slips. We imaged the pancreas and neighbouring mesenchyme of every zebrafish sample that we had mounted with the confocal laser scanning microscopy platform Leica TCS SP8 and LAS X (Leica Microsystems). We analysed the images by Fiji (*Schindelin et al., 2012*) and classified a β-cell as pancreatic when it was located in the pancreas, as delineated by pan-cadherin, DAPI, or *ptf1a:EGFP* labelling. The insulin-positive cells outside of the pancreas were defined as ectopic β-cells.

## Live calcium imaging of zebrafish β-cells

Imaging of pancreatic and ectopic β-cells was performed on 3 dpf *Tg(ins:GCaMP6s);Tg(ins: mCherry);Tg(ins:Flag-NTR) npas4l* mutants on a ZEISS LSM 980 confocal microscope equipped with a W Plan-Apochromat ×20/1 NA water correction lens and operated by ZEN. The GCaMP6s and mCherry signals from β-cells were simultaneously acquired using the 488 nm and 587 nm laser lines. The GCaMP6s signal was rendered in green, and the mCherry signal was rendered in red. Time series recordings were taken with an in-plane resolution of 1024 × 1024 pixels and a fully open pinhole to maximise light capture. The videos were recorded for 400 cycles, with approximately 2 s acquisition time per frame. Videos were analysed in ImageJ (*Schneider et al., 2012*).

## Whole-mount in situ hybridisation

Zebrafish embryos at 10 and 20 dpf were fixed with 4% paraformaldehyde in PBS at 4°C overnight. Whole-mount in situ hybridisation was performed according to the method in a previous report (*Thisse and Thisse, 2008*). Probes against *npas4l* and *drl* were synthesised from transcription templates from PCR using bud-stage zebrafish cDNA, Phusion High-Fidelity DNA Polymerase (Thermo Scientific) and primer pairs 5'-ACTCGGGCATCAGGAGGATC-3' plus 5'-(CCTAATACGACTCACTA TAGGG) GACACCAGCATACGACACACAAC-3' for *npas4l*, and 5'-ATGAAGAATACAACAAAACCC-3' plus 5'-(CCTAATACGACTCACTATAGGG) TGAGAAGCTCTGGCCGC-3' for *drl*, respectively. Plasmids carrying probe templates of *foxa2*, *gsc*, *mixl1*, and *ascl1b* were linearised by corresponding restriction enzymes. T7 (except *foxa2* probe with T3) was employed for transcription, and digoxigenin (Roche) was used for labelling. To genotype the *npas4l^s5* mutants, PCR was performed using gDNA from the imaged samples and primers 5'-TTCCATCTTCTGAATCCTCCA-3' plus 5'-GGACA-GACCCAGATACTCGT-3' at the conditions previously reported (*Reischauer et al., 2016*). The PCR products were then sent for sequencing with the following primer 5'-TTTCTGCCGTGAATGGATG TG-3' (Eurofins Genomics).

## Statistical analysis

Similar experiments were performed at least twice independently. The number of cells in the confocal microscopy images was all quantified manually with the aid of the Multipoint Tool from Fiji (*Schindelin et al., 2012*). Data were then analysed with Prism (GraphPad). Statistical analyses were carried out by two-tailed t-tests when two groups were analysed and by ANOVA when more than two groups were analysed. We have presented the results as the mean values ± SEM and considered p-values ≤ 0.05 to be statistically significant. The n number represents the number of zebrafish individuals in each group of each experiment.

## Acknowledgements

We are very thankful to Abdeljabbar El Manira for making equipment available for calcium imaging; Dirk Meyer for plasmids carrying probe templates of *foxa2*, *gsc*, *mixl1*, and *ascl1b*; Christian Mosimann for sharing the *Tg(−6.35drl:Cre-ERT2,cryaa:Venus)^{cz3333}* line and information; and Naoki Tsuji for the pDestTol2gY plasmid.

## Additional information

### Competing interests

Didier YR Stainier: Senior editor, *eLife*. The other authors declare that no competing interests exist.

### Funding

| Funder | Grant reference number | Author |
| --- | --- | --- |
| Ragnar Söderbergs stiftelse | | Olov Andersson |
| Max Planck Society | | Didier YR Stainier |
| Vetenskapsrådet | | Olov Andersson |
| Novo Nordisk Fonden | | Olov Andersson |
| Karolinska Institutetvia SRP Diabetes & StratRegen | | Olov Andersson |
| H2020 European Research Council | 772365 | Olov Andersson |

The funders had no role in study design, data collection and interpretation, or the decision to submit the work for publication.

## Author contributions
Ka-Cheuk Liu, Data curation, Formal analysis, Investigation, Visualization, Methodology, Writing - original draft, Writing - review and editing; Alethia Villasenor, Formal analysis, Validation, Investigation, Visualization, Writing - review and editing; Maria Bertuzzi, Formal analysis, Investigation, Methodology, Writing - review and editing; Nicole Schmitner, Formal analysis, Investigation, Visualization, Methodology; Niki Radros, Linn Rautio, Formal analysis, Investigation; Kenny Mattonet, Resources, Investigation, Methodology, Writing - review and editing; Ryota L Matsuoka, Resources, Methodology, Writing - review and editing; Sven Reischauer, Investigation, Methodology, Writing - review and editing; Didier YR Stainier, Supervision, Funding acquisition, Investigation, Methodology, Project administration, Writing - review and editing; Olov Andersson, Conceptualization, Formal analysis, Supervision, Funding acquisition, Validation, Investigation, Visualization, Writing - original draft, Project administration, Writing - review and editing

## Author ORCIDs
Ka-Cheuk Liu (iD) https://orcid.org/0000-0001-6082-8801
Kenny Mattonet (iD) http://orcid.org/0000-0002-9705-8086
Ryota L Matsuoka (iD) http://orcid.org/0000-0001-6214-2889
Sven Reischauer (iD) http://orcid.org/0000-0002-6955-9481
Didier YR Stainier (iD) http://orcid.org/0000-0002-0382-0026
Olov Andersson (iD) https://orcid.org/0000-0001-6715-781X

## Ethics
Animal experimentation: This study was performed in accordance with the recommendations of Karolinska Institutet. All of the animals were handled according to approved institutional animal care and in accordance with rules in L150 from the Swedish Board of Agriculture. The protocol was approved by Stockholms Djurförsöksetiska nämnd (Permit Number: N145/15 and 6848-2020).

## Decision letter and Author response
Decision letter https://doi.org/10.7554/eLife.65758.sa1
Author response https://doi.org/10.7554/eLife.65758.sa2

# Additional files

## Supplementary files
• Source data 1. Source data for all Figures.
• Transparent reporting form

## Data availability
All data generated or analysed during this study are included in the manuscript and source data files.

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
