## [Decision Letter]

**Acceptance summary:**

This is an elegant study demonstrating the emergence of mesoderm-derived β-like cells following β-cell ablation in the context of compromised endothelial/myeloid lineage specification. These findings will be of interest to scientists in the areas of regeneration and reprogramming, as they reveal a previously unknown degree of germ layer plasticity in the embryo. In the long term the study has potential impact in the diabetes field, as it reveals a potentially novel path for redirecting somatic cells into insulin-producing cells in an in vivo context.

**Decision letter after peer review:**

Thank you for submitting your article "Regenerating insulin-producing β-cells ectopically from a mesodermal origin in the absence of endothelial specification" for consideration by *eLife*. Your article has been reviewed by 3 peer reviewers, including Elke Ober as the Reviewing Editor and Reviewer #1, and the evaluation has been overseen by Marianne Bronner as the Senior Editor. The following individual involved in review of your submission has agreed to reveal their identity: Valerie Gouon-Evans (Reviewer #3).

Essential Revisions:

1) Improve characterization of β-cell differentiation derived from mesoderm-endoderm conversion. The manuscript provides strong evidence that mesoderm-derived ectopic β-cells generated after β-cell ablation express insulin, nevertheless the characterization is too superficial to conclude there is a full conversion to endoderm. Indeed, the authors describe that in contrast to pancreatic β cells, most ectopic β-cells lack the pan-endocrine marker neurod1 and the key β cell gene pdx1 (Figure 2 and Figure 2-S1), suggesting these cells clearly differ from bona fide β-cells. In addition, analysis of endodermal or mesodermal markers is lacking, making it difficult to understand the exact nature of these ectopic cells. Therefore, a more comprehensive analysis of relevant key endodermal and mesodermal genes is required to better describe the β-cell state of these ectopic cells and determine the degree of germ layer conversion.

2) Address the competence of mesodermal cells converting into β-like cells in larvae with impaired npas4l and etv2 function. The manuscript does not sufficiently discuss the identity of the mesodermal cell population that in the absence of endothelial specification (npas4l-/- and etv2 MO) can adopt a β-cell like identity following β-cell ablation. Specifically, the following points should be discussed:

i) why are they competent to differentiate into insulin+ cells in response to β cell ablation? The authors could make use of the recent studies cited in the present paper and of the respective transcriptomic data (Marass, Development, 2019; Chestnut, Nat Commun, 2020).

ii) is there a temporal window of competence for extra β-cells to form in the mesenchyme? Is this restricted to continued development? Could this be tested in adults?

iii) is the competence to form extra β-cells restricted to the vicinity of the pancreas? If so, which signals control this, given that normally endothelial cells form along the anteroposterior axis, however do not adopt a β-cell like fate in npas4l mutants or etv2morphants?

iv) what is the possible mechanism of this specification?

3) The mesodermal lineage tracing experiments, which are a central part of the study, require additional support. Please define efficiency and specificity of the tracing approach by showing mCherry+ mesodermal cells co-stained with a mesodermal marker (e.g. drl, or alternative mesodermal gene expressed at the stages of ectopic β-cell analysis) in additional areas of the larvae. This should include control embryos for both mesodermal Cre-drivers (drl:CreERT2 and etv2:iCre) to show their exclusive mesoderm expression.

4) The etv2 knockdown requires further validation. Elongated cells resembling endothelial cells marked with the etv2 lineage mapping present in etv2 morphants suggests that endothelial cells still form. Related to this, using only a single etv2 and single standard control morpholino is surprising given the known problems with this approach. Please validate the use of the etv2 morpholino knockdowns according to Stainier et al., 2017. Moreover, endothelial cell identity of above cells should be tested.

In case these results are confirmed, it would suggest that the mechanism of ectopic β-cells emergence in etv2 morphants and npas4l mutants does not depend on the absence of endothelial cells per se. It may also suggest that the mechanism of ectopic β cells formation in etv2 morphants vs npas4l mutant is different. This should be discussed as the role of vasculogenesis and vascularization for β cell regeneration was the first motivation for this study, as introduced in the result section.

5) Clarification of assessment criteria for different insulin+ cell populations and their localization to the principle islet or ectopic tissues.

The regenerated β cells seem to be broadly distributed and not always seem to form a distinct islet, but are nevertheless assigned to the pancreatic domain (see Figure 1E). To clarify how the pancreatic and the ectopic domains have been discriminated in these experiments, please provide additional details how both populations have been identified.

Moreover, lineage labelled Ins+Cherry+ cells from the mesoderm are also found within the pancreas; for instance a substantial fraction of regenerated β-cells defined as pancreatic seems to derive from etv2+ mesodermal cells in etv2 morphants (Figure 4B). This raises the question of the identity and of these "pancreatic" cells and where they arise from, and requires additional characterization and explanation.

The β-like cells appear to come from the lateral plate mesoderm (supplement figure4), please specify at least once in the text for clarity of lineage relationship.

6) The interpretation of glucose measurements needs to be revisited, given the relevance of measuring free tissue glucose as proxy for the effect of insulin released by ectopic cells in npas4l mutants is questionable. npas4l mutants have no capillaries nor blood to vehicle the hormone to target tissues in mutants. It is likely that the glucose values obtained for the mutants are not the result of insulin action, and hence may not provide a reliable estimation of ectopic cell function. This would be more convincing, for example, if a transient increase is observed after ablation before normalization. If this is the case, the authors should experimentally test whether increasing glucose levels in non-ablated npas4l mutants would trigger a similar formation of ectopic β-cells.

Therefore, please provide further support or adjust and discuss the conclusions.

7) The significance of the conversion of other endocrine cell types, specifically somatostatin+ cells, from mesoderm in npas4l mutants needs to be clarified. The increase in the number of ectopic sst2 cells after δ-cell ablation presented in Figure 1 Supplement 2F is very mild and in addition ectopic sst2 cell seem to frequently form in controls outside the pancreas. Please clarify and discuss general implications for the competence of the mesenchyme for the appearance of endocrine cells outside the pancreas, as in its current form, this result is more confusing than supporting the other results.

Has the normal distribution of data points in Figure 1 supplement 2F been tested for control and npas4l datasets?

8) A number of clarifications and additions to the text and figures are required:

- The title requires attention, it seems grammatically incorrect, e.g. lacking a verb.

- Figure 2 indicates that the extra b-cell phenotype is not fully penetrant, please include this in the manuscript.

- Clarify and specify in all figure legends what the red staining represents, e.g. DsRed fluorescence driven by the insulin promoter or immunofluorescence staining for insulin?

- Figure 1, supplement 1: Please delineate the pancreas in panels C and D to reveal the ectopic sites of emerging β-like cells.

- p-Cadherin: please spell out PanCadherin in the figures to avoid confusion with a phosphorylated Cadherin form.

- Please include short explanation for iCre in the text or methods.

[Editors' note: further revisions were suggested prior to acceptance, as described below.]

Thank you for resubmitting your work entitled "Insulin-producing β-cells regenerate ectopically from a mesodermal origin under the perturbation of hemato-endothelial specification" for further consideration by *eLife*. Your article has been reviewed by 2 peer reviewers, and the evaluation has been overseen by a Reviewing Editor and Marianne Bronner as the Senior Editor. The following individual involved in review of your submission has agreed to reveal their identity: Valerie Gouon-Evans (Reviewer #3).

The manuscript has been improved but there are some remaining issues that need to be addressed, as outlined below.

The manuscript does not require any further experimental data, nevertheless, we ask you to address some important additions to the text, as suggested by the reviewers. Including the following changes will add essential information and will help to make your work more accessible to the reader:

– mention once in the manuscript the previous name of etsrp (etv2) for the sake of visibility and clarity.

– precise when 4OHT was applied in the control experiment shown in Figure 4 – Supplement figure 3 (drl:CRE-ERT in the heart).

– include the characterization of specificity and efficiency of the lineage tracing etsrp:Cre in the manuscript as described in the rebuttal letter (response to point 3).

– summarize lines 289-294 (Discussion) to simply suggest that the possibly versatile mutant mesodermal cells could be the origin of the ectopic β cells.

*Reviewer #2 (Recommendations for the authors):*

The revised version of this manuscript and the rebuttal letter carefully address all the concerns and the remarks raised by the first version:

The title and the general conclusions have been adequately adapted so that they more fairly correspond to the observed phenomenon of ectopic pancreatic β cell formation in the mesoderm. The manuscript now presents some interesting discussion about the plasticity of mesodermal cells.

Importantly, the authors have better characterized their lineage tracing tools. Also, they include a third tool, a GAL4-VP16 knockin in the npas4l locus to be used with UAS:CRE;ubb:switch. Together, these three lineage tracing systems convincingly support the conclusion that many ectopic pancreatic cells are from mesodermal origin. It is to note, however, that this third tool is only minimally described as it is said it will be presented in another paper submitted by colleagues.

The study now provides in vivo Calcium imaging of ectopic (compared to pancreatic) β cells, which is an accurate way to assess the capacity of these cells to respond to glucose. This also adds to the asked characterization of the ectopic β cells.

The use of a secondo morpholino against etv2/etsrp is now presented, as well as the synergy between both (with respect to the number of ectopic β cells that form during regeneration). This analysis complies with previously published guidelines for MO.

In addition, the figures and Methods provide clarification about the way cells have been assigned a pancreatic or ectopic localization.

Source data for the quantifications shown in each figure are provided.

Overall, the revised version is greatly improved.

Additional recommendations:

– mention once in the manuscript the previous name of etsrp (etv2) for the sake of visibility and clarity.

– precise when 4OHT was applied in the control experiment shown in Figure 4 – Supplement figure 3 (drl:CRE-ERT in the heart).

– summarize lines 289-294 (Discussion) to simply suggest that the possibly versatile mutant mesodermal cells could be the origin of the ectopic β cells.

*Reviewer #3 (Recommendations for the authors):*

Overall, the revisions are satisfactory and have greatly improved the manuscript.

Point #1: I agree that the new data using the npas4lPt(+36-npas4l-p2a-Gal4-VP16)bns423 fish crossed either with Tg(UAS:Cre);Tg(ubi:Switch) or Tg(UAS:EGFP) fish provide additional evidence for the mesodermal origin of the de novo β cells that are characterized with loss of the mesodermal marker expression npas4l.

Additional discussion addresses well point #2.

Point #3: Please include the characterization of specificity and efficiency of the lineage tracing etsrp:Cre in the manuscript as described in the rebuttal letter.

Point #4: MO2 data support well the previous MO1 data.

Point #5 is well addressed.

Point #6: Evaluation of calcium signaling in the ectopic β-cells by live imaging in vivo is useful to examine cell maturation.

Point #7: Removal of data related to somatostatin+ cells is appropriate.

Point #8: Well addressed.

---

## [Author Response]

Essential Revisions:1) Improve characterization of β-cell differentiation derived from mesoderm-endoderm conversion. The manuscript provides strong evidence that mesoderm-derived ectopic β-cells generated after β-cell ablation express insulin, nevertheless the characterization is too superficial to conclude there is a full conversion to endoderm. Indeed, the authors describe that in contrast to pancreatic β cells, most ectopic β-cells lack the pan-endocrine marker neurod1 and the key β cell gene pdx1 (Figure 2 and Figure 2-S1), suggesting these cells clearly differ from bona fide β-cells. In addition, analysis of endodermal or mesodermal markers is lacking, making it difficult to understand the exact nature of these ectopic cells. Therefore, a more comprehensive analysis of relevant key endodermal and mesodermal genes is required to better describe the β-cell state of these ectopic cells and determine the degree of germ layer conversion.

We agree with the reviewers that the ectopic β-cells are not identical to the *bona fide* β-cells and more characterization of the ectopic β-cells would be necessary to fully understand their cellular state. Therefore in the revision we have included a new zebrafish tool *npas4l^Pt(+36-npas4l-p2a-Gal4-VP16)bns423^*, which not only allows us to trace the mesodermal *npas4l*-positive lineage of the endothelial and hematopoietic progenitors after crossing into *Tg(UAS:Cre);Tg(ubi:Switch)*, but also reports the expression of this important regulator of the mesodermal cell fate after crossing into *Tg(UAS:EGFP)*. The details and characterization of this new fish line are reported in another manuscript by K.M. and D.Y.R.S (submitted).

In the new Figure 5, we have shown that some of the ectopic β-cells could be traced back to the mesodermal *npas4l* lineage while none of the ectopic β-cells were still actively expressing *npas4l* as reported by EGFP at 3 dpf. This indicates that at least some of these ectopic β-cells from the *npas4l* lineage had lost this mesodermal marker and begun to express insulin and other endodermal endocrine markers (Figure 2).

In addition, we examined the expression patterns of some early mesodermal or endodermal markers (*foxa2*, *gsc* and *mixl1*) and we did not see any significant differences in the *npas4l* mutants (Figure 4—figure supplement 1), indicating that *npas4l* mutation did not significantly perturb the early formation of mesoderm and endoderm.

2) Address the competence of mesodermal cells converting into β-like cells in larvae with impaired npas4l and etv2 function. The manuscript does not sufficiently discuss the identity of the mesodermal cell population that in the absence of endothelial specification (npas4l-/- and etv2 MO) can adopt a β-cell like identity following β-cell ablation. Specifically, the following points should be discussed:i) why are they competent to differentiate into insulin+ cells in response to β cell ablation? The authors could make use of the recent studies cited in the present paper and of the respective transcriptomic data (Marass, Development, 2019; Chestnut, Nat Commun, 2020).

We thank the reviewers’ suggestion of making use of these two studies to discuss the competence of the mesodermal cell population differentiating into ectopic β-cells. We have now substantially expanded this discussion point and referred to the suggested references (Lines 277-305).

ii) is there a temporal window of competence for extra β-cells to form in the mesenchyme? Is this restricted to continued development? Could this be tested in adults?

The formation of ectopic β-cells, unfortunately, cannot be tested in adult *cloche* mutants because their health conditions start to deteriorate from 3 days post-fertilisation (dpf) and they die at 4-5 dpf. Therefore, they can never reach the adult stage. We analysed the cloche mutants at 3 dpf in the current manuscript but we also tried to examine them at 4 dpf, but there were worsened health conditions of the *cloche* mutants in general that prohibited a meaningful analysis. In addition, the first insulin-expressing β-cell does not show up before 15 hr post-fertilisation (hpf). Thus, there is not much room for us to move the ablation procedure earlier than what we did in the current manuscript (with a start at ~24 hpf).

iii) is the competence to form extra β-cells restricted to the vicinity of the pancreas? If so, which signals control this, given that normally endothelial cells form along the anteroposterior axis, however do not adopt a β-cell like fate in npas4l mutants or etv2morphants?

We believe that the vicinity of the pancreas could be an important factor for the competence of some mesodermal cells to form ectopic β-cells. However, this is difficult to prove before we precisely know which population of mesodermal cells can differentiate across the germ layers to endodermal β-cells. If we had knowledge of this, we could have isolated those cells and transplanted them into different positions to study the importance of the vicinity of the pancreas.

In addition, our new lineage-tracing data has shown that some of the ectopic β-cells could be traced back to the *npas4l* lineage (Figure 5), suggesting that the *npas4l*-expressing lateral plate mesoderm not far from the pancreas (Figure 4—figure supplement 1) could be the source of the ectopic β-cells. We have added this point to the newly expanded discussion (Line 296).

iv) what is the possible mechanism of this specification?

As supported by the transcriptomic data (Marass, Development, 2019; Chestnut, Nat Commun, 2020), we propose that a small population of cells from the lateral plate mesoderm have lost their mesodermal cell fate in *npas4l* mutants or *etsrp* (*etv2*) morphants, which maintains the plasticity of these cells and allows them to differentiate into ectopic β-cells during β-cell regeneration. We have discussed this possible mechanism in the expanded discussion (Lines 277-305), although the detailed molecular mechanisms remain elusive.

3) The mesodermal lineage tracing experiments, which are a central part of the study, require additional support. Please define efficiency and specificity of the tracing approach by showing mCherry+ mesodermal cells co-stained with a mesodermal marker (e.g. drl, or alternative mesodermal gene expressed at the stages of ectopic β-cell analysis) in additional areas of the larvae. This should include control embryos for both mesodermal Cre-drivers (drl:CreERT2 and etv2:iCre) to show their exclusive mesoderm expression.

We thank the reviewers for pointing out the importance of characterizing the Cre-drivers for lineage-tracing, especially with regards to the *etsrp:Cre* (which we previously called *etv2:iCre*), which has been newly made for this manuscript. We understand the significance of having an efficient and specific lineage-tracing approach, while we also realize that no approach is absolutely perfect. Therefore, in this revised manuscript, we have included yet another lineage tracer *npas4l^Pt(+36-npas4l-p2a-Gal4-VP16)bns423^;UAS:Cre;ubi:Switch.* Together with the other two mesodermal lineage tracers *drl:CreER^T2^;ubi:Switch* and *etsrp:Cre;ubi:Switch*, we hope these three lineage tracers would complement each other to provide convincing evidence to show that the ectopic β-cells came from an mesodermal origin.

The Cre-driver *drl:CreER^T2^
*has been extensively characterized by Mosimann and others [Nat Commun 6, 8146 (2015)] and it can genetically trace the lateral plate mesoderm descendants including cardiac mesoderm, pectoral find mesoderm, red blood cells, kidney, vasculature and trunk muscle cells. In the revised manuscript, we have added images to show that this tracer could respond to tamoxifen induction and genetically trace cardiac cells (Figure 4—figure supplement 3). Moreover, we have constructed *etsrp:Cre* based on the promoter of *-2.3estrp:GFP*, which has been demonstrated to closely recapitulate the endogenous *etsrp* expression in lateral plate mesoderm and vasculatures by Veldman and Lin [Circ Res 110, 220-9 (2012)]. We have shown that *etsrp:Cre* can trace over 80% of *kdrl:GFP*-expressing intersegmental vessels (Figure 6—figure supplement 1). Furthermore, the new fish model *npas4l^Pt(+36-npas4l-p2a-Gal4-VP16)bns423^* has been characterised in another manuscript by K.M. and D.Y.R.S (submitted).

4) The etv2 knockdown requires further validation. Elongated cells resembling endothelial cells marked with the etv2 lineage mapping present in etv2 morphants suggests that endothelial cells still form. Related to this, using only a single etv2 and single standard control morpholino is surprising given the known problems with this approach. Please validate the use of the etv2 morpholino knockdowns according to Stainier et al., 2017. Moreover, endothelial cell identity of above cells should be tested.In case these results are confirmed, it would suggest that the mechanism of ectopic β-cells emergence in etv2 morphants and npas4l mutants does not depend on the absence of endothelial cells per se. It may also suggest that the mechanism of ectopic β cells formation in etv2 morphants vs npas4l mutant is different. This should be discussed as the role of vasculogenesis and vascularization for β cell regeneration was the first motivation for this study, as introduced in the result section.

We agree with the reviewers that the formation of ectopic β-cells does not require the complete absence of endothelial cells and it is common that the phenotypes in morphants does not have complete penetrance. Sumanas and Lin have also shown the dose-dependent and incomplete effects of *etsrp* (*etv2*) morpholinos [PLoS Biol 4(1):e10 (2006)]. Therefore, we have changed the title of the manuscript to “Insulin-producing β-cells regenerate ectopically from a mesodermal origin under the perturbation of hemato-endothelial specification”.

In the revision, we have added another widely used *etsrp* morpholino (MO1) in addition to the MO2 included in the original submission. Both MO1 and MO2 could induce the formation of ectopic β-cells independently compared to the random control morpholino (instead of the standard control). Importantly, co-injecting lower doses of MO1 and MO2 could exert a synergistic effect on ectopic β-cell formation (Figure 4—figure supplement 4), suggesting that both morpholinos are specifically targeting *etsrp* (in line with the suggestions in Stainier et al., 2017). Interestingly, similar to the higher effectiveness of MO2 on inducing the complete loss of circulation reported by Sumanas and Lin [PLoS Biol 4(1):e10 (2006)], MO2 also tended to be more potent than MO1 in inducing ectopic β-cell formation. We have also tried to block the ectopic β-cell formation by co-injecting *etsrp* mRNA with MO2. However, *etsrp* mRNA did not significantly stop the ectopic β-cell formation, which might be due to mRNA degradation.

It is true that we cannot absolutely rule out the possibility that the mechanisms of ectopic β-cell formation in *npas4l* mutants and *etsrp* morphants could be different. Therefore, we have added a statement in the discussion part (Lines 302-305). However, the lack of vasculatures is unlikely the major driving force for ectopic β-cell formation because we did not observe the same phenotype in zebrafish embryos treated with chemicals that inhibit angiogenesis.

5) Clarification of assessment criteria for different insulin+ cell populations and their localization to the principle islet or ectopic tissues.The regenerated β cells seem to be broadly distributed and not always seem to form a distinct islet, but are nevertheless assigned to the pancreatic domain (see Figure 1E). To clarify how the pancreatic and the ectopic domains have been discriminated in these experiments, please provide additional details how both populations have been identified.Moreover, lineage labelled Ins+Cherry+ cells from the mesoderm are also found within the pancreas; for instance a substantial fraction of regenerated β-cells defined as pancreatic seems to derive from etv2+ mesodermal cells in etv2 morphants (Figure 4B). This raises the question of the identity and of these "pancreatic" cells and where they arise from, and requires additional characterization and explanation.The β-like cells appear to come from the lateral plate mesoderm (supplement figure4), please specify at least once in the text for clarity of lineage relationship.

We thank the reviewers for expressing concerns about the criteria for defining the pancreatic and ectopic β-cells. Now we have clarified in the Methods part how we classified these two domains during the various analyses (Lines 443-447). It was completely based on the location of the β-cells—i.e., whether they were in the pancreas or not, while the region of the pancreas was defined by the signals of *ptf1a:EGFP*, DAPI or pan-cadherin staining.

We agree that the ectopic β-cells were not very clearly illustrated in the previous Figure 1E when we tried to keep the dashed rectangle intact. In this revised version, we have modified the way of illustration and delineated the whole pancreatic domains to make things clearer (Figure 1D and E). Newly regenerated β-cells usually appear more dispersed than those under basal development without ablation and it often takes a few more days of recovery before they cluster together.

Without the inducible ER^T2^ temporal control, some low- or miss-expression of *etsrp:Cre* (*etv2:iCre*) in early progenitors from mesendoderm might be sufficient enough to trigger the Cre-recombination *ubi:Switch* and genetically traced a few descendants of mesendoderm, including the endodermal pancreatic islet. Therefore, in this revised manuscript, we have included three different lineage-tracing models to complement each other to minimise the influence of any confounding effect.

In addition, we have added the point of the lateral plate mesoderm being a potential origin of the ectopic β-cells in the expanded discussion (Lines 277-305).

6) The interpretation of glucose measurements needs to be revisited, given the relevance of measuring free tissue glucose as proxy for the effect of insulin released by ectopic cells in npas4l mutants is questionable. npas4l mutants have no capillaries nor blood to vehicle the hormone to target tissues in mutants. It is likely that the glucose values obtained for the mutants are not the result of insulin action, and hence may not provide a reliable estimation of ectopic cell function. This would be more convincing, for example, if a transient increase is observed after ablation before normalization. If this is the case, the authors should experimentally test whether increasing glucose levels in non-ablated npas4l mutants would trigger a similar formation of ectopic β-cells.Therefore, please provide further support or adjust and discuss the conclusions.

We thank the reviewers for pointing out the potential problems of the glucose experiment. Hence, we tried to collect the zebrafish samples earlier at different time points to see if there was a transient increase in free glucose levels in the *npas4l* mutants after β-cell ablation. However, we only sometimes saw a mild but insignificant increase in the glucose levels, suggesting that there was likely no severely elevated glucose level for the ectopic β-cells to tackle with and increased glucose level was not a crucial factor for the formation of the ectopic β-cells.

Instead, during the revision of the manuscript, we have examined the calcium signalling in the ectopic β-cells by live imaging in vivo to see how mature they are as secretory cells. Similar to some of the pancreatic β-cells, a portion of ectopic β-cells also displayed certain basal calcium activity at the baseline and responded to glucose treatments with calcium oscillations in the *npas4l* mutants (Figure 3), suggesting that some of the ectopic β-cells are glucose-responsive secretory cells.

7) The significance of the conversion of other endocrine cell types, specifically somatostatin+ cells, from mesoderm in npas4l mutants needs to be clarified. The increase in the number of ectopic sst2 cells after δ-cell ablation presented in Figure 1 Supplement 2F is very mild and in addition ectopic sst2 cell seem to frequently form in controls outside the pancreas. Please clarify and discuss general implications for the competence of the mesenchyme for the appearance of endocrine cells outside the pancreas, as in its current form, this result is more confusing than supporting the other results.Has the normal distribution of data points in Figure 1 supplement 2F been tested for control and npas4l datasets?

We agree with the reviewers that the data of somatostatin+ cells may not be very helpful. Therefore, we have removed this part of the data from the revised manuscript.

8) A number of clarifications and additions to the text and figures are required:

– *The title requires attention, it seems grammatically incorrect, e.g. lacking a verb.*

We have changed the title to “Insulin-producing β-cells regenerate ectopically from a mesodermal origin under the perturbation of hemato-endothelial specification”.

– *Figure 2 indicates that the extra b-cell phenotype is not fully penetrant, please include this in the manuscript.*

The quantifications of Figure 2 are not optimal for describing the penetrance of the phenotype because an individual with all the Ins+ cells carrying a marker (e.g. Ins+Isl1+ cells) would result in zero Ins+ only cells in the same individual but it does not mean the individual carries no ectopic Ins+ cells. Figure 1 is more suitable for showing the penetrance with the total ectopic β-cell number. We did not get 100% penetrance in every single experiment (although the penetrance of ectopic β-cells is very high), which could be dependent on different β-cell reporters/immunostaining, slight variations of developmental stages of individuals, or mild variations of the β-cell ablation.

– *Clarify and specify in all figure legends what the red staining represents, e.g. DsRed fluorescence driven by the insulin promoter or immunofluorescence staining for insulin?*

We thank the reviewers’ suggestion and we have clarified the red fluorescence in the figure legends in the revised manuscript.

– *Figure 1, supplement 1: Please delineate the pancreas in panels C and D to reveal the ectopic sites of emerging β-like cells.*

We have added white lines to delineate the pancreas in Figure 1—figure supplement 1.

– *p-Cadherin: please spell out PanCadherin in the figures to avoid confusion with a phosphorylated Cadherin form.*

We have changed the term p-Cadherin to pan-Cadherin.

– *Please include short explanation for iCre in the text or methods.*

The iCre used in the study is improved Cre-recombinase. We have abbreviated iCre as Cre to avoid confusion and added a short description of it in the methods (Lines 364-366).

[Editors' note: further revisions were suggested prior to acceptance, as described below.]

The manuscript does not require any further experimental data, nevertheless, we ask you to address some important additions to the text, as suggested by the reviewers. Including the following changes will add essential information and will help to make your work more accessible to the reader:– mention once in the manuscript the previous name of etsrp (etv2) for the sake of visibility and clarity.

Now we have mentioned the previous name of *etsrp* in the introduction (line 78 in the word document / line 79 after conversion to the pdf document).

– precise when 4OHT was applied in the control experiment shown in Figure 4 – Supplement figure 3 (drl:CRE-ERT in the heart).

The information has been added to the figure legend (line 946 in word/961 in pdf).

– include the characterization of specificity and efficiency of the lineage tracing etsrp:Cre in the manuscript as described in the rebuttal letter (response to point 3).

The description of the characterisation has been further expanded (line 229-231 in word/234-236 in pdf).

– summarize lines 289-294 (Discussion) to simply suggest that the possibly versatile mutant mesodermal cells could be the origin of the ectopic β cells.

Text changes have been made to make a simpler summary (line 291-293/298-300 in pdf).